# Improving the LPJmL4-SPITFIRE vegetation-fire model for South America using satellite data

Markus Drüke[1,2], Matthias Forkel[3], Werner von Bloh[1], Boris Sakschewski[1], Manoel Cardoso[4], Mercedes Bustamante[5], Jürgen Kurths[1,2], and Kirsten Thonicke[1]

[1]Potsdam Institute for Climate Impact Research, Telegraphenberg A 31, Potsdam, Germany
[2]Humboldt Universität zu Berlin, Unter den Linden 6, 10099 Berlin, Deutschland
[3] TU Wien, Department of Geodesy and Geoinformation, Gusshausstr. 27-29, 1040 Vienna, Austria
[4]Instituto Nacional de Pesquisas Espaciais, Av. dos Astronautas, 1.758 - Jardim da Granja, São José dos Campos - SP, 12227-010, Brazil
[5]Instituto de Ciências Biologicas, Universidade de Brasília, Campus Universitário Darcy Ribeiro - Asa Norte, 70910-900 Brasília, Brazil

**Correspondence:** Markus Drüke (drueke@pik-potsdam.de)

**Abstract.** Vegetation fires influence global vegetation distribution, ecosystem functioning, and global carbon cycling. Specifically in South America, changes in fire occurrence together with land use change accelerate ecosystem fragmentation and increase the vulnerability of tropical forests and savannas to climate change. Dynamic Global Vegetation Models (DGVMs) are valuable tools to estimate the effects of fire on ecosystem functioning and carbon cycling under future climate changes. However, most fire-enabled DGVMs have problems in capturing the magnitude, spatial patterns, and temporal dynamics of burnt area as observed by satellites. As fire is controlled by the interplay of weather conditions, vegetation properties and human activities, fire modules in DGVMs can be improved in various aspects. In this study we focus on improving the controls of climate and hence fuel moisture content on fire danger in the LPJmL4-SPITFIRE DGVM in South America and especially for the Brazilian fire-prone biomes Caatinga and Cerrado. We therefore test two alternative model formulations (standard Nesterov index and a newly implemented water vapor pressure deficit) for climate effects on fire danger within a formal model-data integration setup where we estimate model parameters against satellite data sets of burnt area (GFED4) and above ground biomass of trees. Our results show that the optimized model improves the representation of spatial patterns and the seasonal to inter-annual dynamics of burnt area especially in the Cerrado/Caatinga region. In addition, the model improves the simulation of above-ground biomass and the spatial distribution of plant functional types (PFTs). We obtained the best results by using the water vapor pressure deficit (VPD) for the calculation of fire danger. The VPD includes, in comparison to the Nesterov index, a representation of the air humidity and the vegetation density. This work shows the successful application of a systematic model-data integration setup, as well as the integration of a new fire danger formulation, in order to optimize a process-based fire-enabled DGVM. It further highlights the potential of this approach to achieve a new level of accuracy in comprehensive global fire modelling and prediction.

# 1 Introduction

Fire in the Earth system is an important disturbance leading to many changes in the vegetation and has substantial impact on biodiversity, human health and ecosystems (Langmann et al., 2009). Fire is responsible for ca. 2 Pg of carbon emissions, which constitutes 20 % of global carbon emissions (Giglio et al., 2013; Werf et al., 2010). Fire-induced aerosol emissions and land surface changes modify evapotranspiration and surface albedo and have therefore a crucial impact on global climate (van der Werf et al., 2008; Yue and Unger, 2018). Despite a tendency for globally declining burnt area (Andela et al., 2017; Forkel et al., 2019b), more frequent and intense drought-periods lead to increasing fire-prone weather and surface conditions worldwide and therefore fire danger (Jolly et al., 2015). Growing fire danger along with land-use change are increasing ecosystem's vulnerability, which could in turn shift entire regions into a less vegetated state (Silvério et al., 2013). To account for these effects, it is extremely important to include well performing fire modules in Dynamic Global Vegetation Models (DGVMs).

Especially in South America, tropical forests, woodlands and other ecosystems are vulnerable to increasing fire danger and land use change (Cochrane and Laurance, 2008). This study focuses on the fire behavior in central-northern South America and especially on the Brazilian biomes Caatinga and Cerrado, which are the most fire-prone regions in South America (Fig. 1). Together with the Amazon rainforest they form an area of very high biodiversity and have a large impact on the global carbon-cycle and the regional water cycle (Lahsen et al., 2016). Compared to the Amazon, the Cerrado and Caatinga are both less

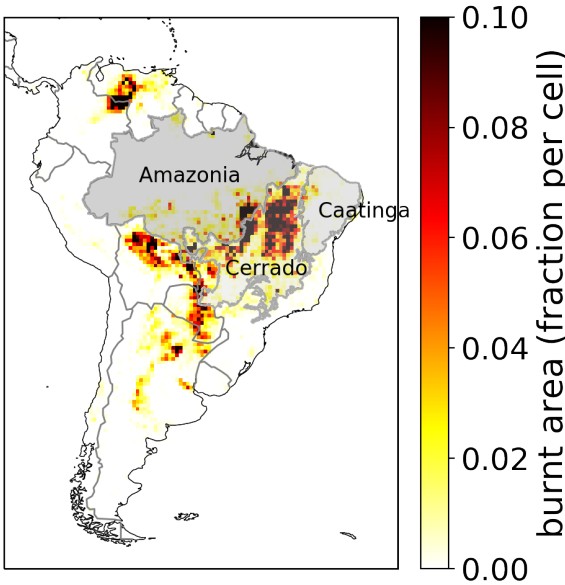

**Figure 1.** Overview of the mean annual burnt area in Brazil from 2005-2015 (Werf et al., 2017; Giglio et al., 2013) and the biomes Amazonia, Cerrado and Caatinga (IBGE, 2019; Harvard, 2019)

densely vegetated and drier biomes, but with very different vegetation and precipitation dynamics. The Cerrado is a savanna-like biome with a mixture of shrubs, high grasses and dry forest parts. With a precipitation of ca. 1500 mm per year the Cerrado

does experience a rainy season. The Caatinga, on the other hand, has a semi-arid climate with irregular rainfall between 500 and 750 mm per year, mostly within only a few months of the year. The vegetation is heterogeneous and characterized by deciduous dry forest and shrubs (Alvares et al., 2013; Prado, 2003). The different vegetation types of the Caatinga and the Cerrado lead to different fire spread, fire intensity, fire resistance and fire mortality properties. While within the Cerrado fire is

a frequent event and the plants are mostly adapted to it (70 % of burnt area in Brazil is within the Cerrado, Moreira de Araújo et al., 2012), the Caatinga has a lower fire intensity and fire spread due to a lower biomass, available for fuel. Such variability in the vegetation and dead fuel composition, within and between biomes, poses a challenge to global fire models to correctly simulate observed fire patterns for a variety of biomes. Both, the Caatinga and the Cerrado depend on a strict equilibrium of fire-vegetation-climate feedbacks (Lasslop et al., 2016), which is threatened to be disturbed by human impact through climate

change and land use change (Beuchle et al., 2015). While the Amazon is the focus of various national and international conservation efforts and at least by law well protected, the Cerrado is currently over-exploited by the agribusiness and its importance for regional climate, biodiversity and the water cycle is often neglected (Lahsen et al., 2016). In particular the disturbance of increasing fire regimes by climate change and land-use change might accelerate biome degradation. These effects on the Cerrado might also impact the Amazon rainforest by shifting the position of the savanna-forest biome boundary towards forest,

putting the functioning of the Amazon rain forest at risk (Chambers and Artaxo, 2017). Parts of the Cerrado are also itself vulnerable to increasing fire regimes, and might shift to a less vegetated state, similar to the Caatinga (Hoffmann et al., 2000). To model these feedback processes and to study the range of biome-stability under certain drought-induced perturbations, a realistic fire representation in climate and vegetation models is essential. However, modelling fire behavior of the Brazilian Cerrado and Caatinga presents a huge challenge.

The fire occurrence depends on many interconnected parameters as humidity, precipitation, temperature, ignition sources (lightning and human) and windspeed, but also on fuel load, fuel moisture and the adaption of plant traits to fire (Keeley et al., 2011), which makes the development of fire models a complex task (Forkel et al., 2019a; Hantson et al., 2016; Lasslop et al., 2015; Krawchuk and Moritz, 2011; Jolly et al., 2015). Global fire modelling is done either by empirical models (e.g. Thonicke et al., 2001; Knorr et al., 2016; Forkel et al., 2017) or by process-based models (e.g. Venevsky et al., 2002; Thonicke et al., 2010). Em-

pirical fire models are simplified statistical representations of fire processes and are based on empirical relationships between variables (e.g. soil moisture and fire occurrence). Process-based fire models attempt to simulate fire via explicit process-based relations: Fire ignitions are calculated by taking into account lightning flashes as natural sources and human ignitions. The chance of an ignition to become a spreading fire is then determined by the fire danger index. Sophisticated fire models calculate the rate of spread by taking into account wind speed and then translate these results into an area burnt, fuel consumption

and fire carbon emissions (e.g. Thonicke et al. (2010); see Hantson et al. (2016) for an overview of global fire models). Weather conditions control the moisture content of fuels and the danger of fire to ignite and spread. Hence the simulation of fire danger plays an important role to simulate the occurrence of fire within global process-based fire models (Pechony and Shindell, 2009). Temperature, precipitation, humidity and vegetation-related variables are often used to compute fire weather indices and hence to estimate the risk of ignitions to become a spreading fire (Chuvieco et al., 2010). Various fire weather

indices are used within operational fire danger assessment systems (e.g. Canadian Fire Weather index, FWI (Wagner et al.,

1987), the Keetch Byram Drought Index (Keetch and Byram, 1968), the Angström Fire Danger Index (Arpaci et al., 2013), and the Nesterov index (Venevsky et al., 2002)). However, regional studies show that fire weather indices tend to have different predictive performances for fire occurrence (Arpaci et al., 2013). Hence, the performance of different fire weather indices should be ideally tested in order to accurately represent fire danger in DGVMs. However, not all fire weather indices can be easily adapted for global fire models because they require input variables that are not available within a DGVM framework. Hence a fire danger index for a DGVM should be as complex as necessary but still relatively easy to implement. As a result, the relatively simple Nesterov index has been widely used within global fire models (Venevsky et al., 2002; Thonicke et al., 2010).

Here, we aim to improve the simulated occurrence of fire (i.e. burnt area) in the LPJmL4-SPITFIRE model for South America and in particular for the fire-prone biomes Cerrado and Caatinga. We aim to evaluate the performance of two alternative fire danger indices within SPITFIRE based on the already implemented Nesterov index (Venevsky et al., 2002) and the newly implemented water vapor pressure deficit (VPD thereafter, Pechony and Shindell, 2009; Ray et al., 2005). Furthermore, we apply a formal model-data integration framework (LPJmLmdi, Forkel et al., 2014) to estimate model parameters that control fire danger, fire behavior, fire resistance and mortality against satellite-based data sets of burnt area and above-ground biomass (Fig. 2). Our approach is likely to improve the representation of spatial-temporal variations in fire behavior in different biomes to enable a much better modelling of the impact of climate change on fire-vegetation interactions in the current century.

## 2 Materials and Methods

### 2.1 The coupled vegetation-fire model LPJmL4-SPITFIRE

#### 2.1.1 LPJmL 4.0

The LPJmL 4.0 model (Lund-Potsdam-Jena managed Land, Schaphoff et al., 2018a, b), is a well established and validated process-based DGVM, which globally simulates the surface energy balance, water fluxes, carbon fluxes and stocks, and natural and managed vegetation from climate and soil input data. LPJmL simulates global vegetation distribution as the fractional coverage of plant functional types (PFT), which is called foliage projective cover (FPC), and managed land as fractional coverage of crop functional types (CFT). The establishment and survival of different PFTs is regulated through bioclimatic limits and effects of heat, productivity and fire on plant mortality. Therefore, it enables LPJmL to investigate feedbacks, for example between vegetation and fire. In standard settings, which are also used here, the model operates on the grid of $0.5° \times 0.5°$ latitude-longitude with a spinup time of 5000 years, repeating the first 30 years of the given climate data set.

Since its original implementation by Sitch et al. (2003), LPJmL has been improved by a representation of the water balance (Gerten et al., 2004), a representation of the agriculture (Bondeau et al., 2007), and new modules for fire (Thonicke et al., 2010), permafrost (Schaphoff et al., 2013) and phenology (Forkel et al., 2014).

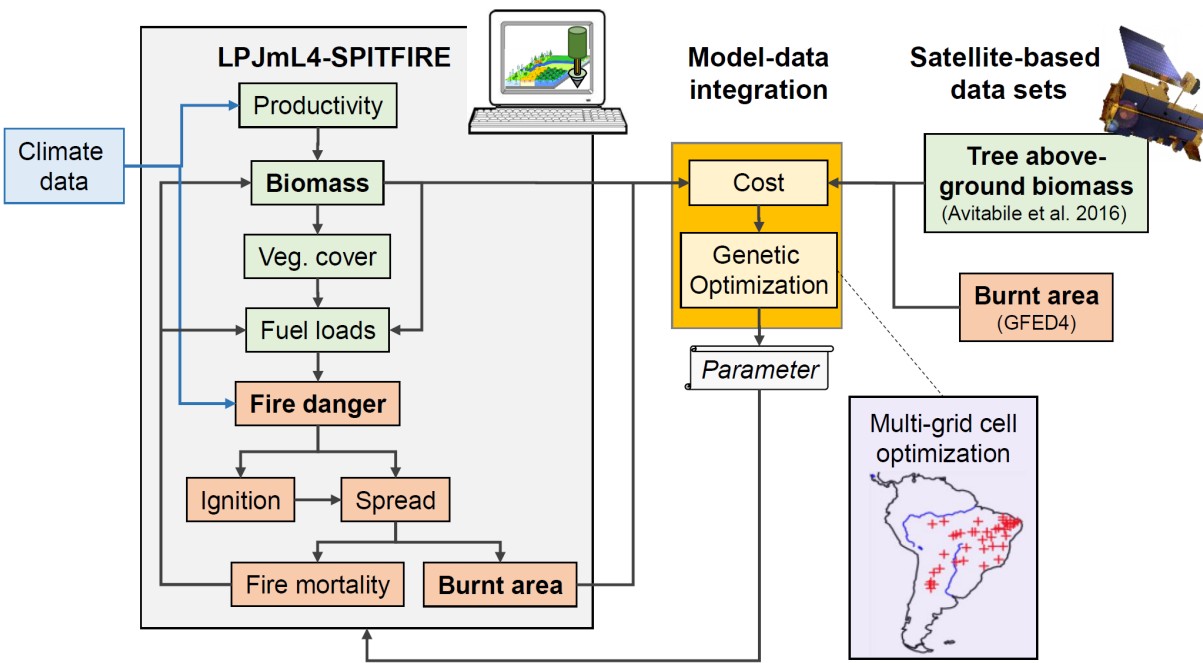

**Figure 2.** Schematic overview of the model-data integration approach to estimate parameters of LPJmL4-SPITFIRE against satellite-based data sets of burnt area and above-ground biomass

### 2.1.2 SPITFIRE

SPITFIRE (SPread and InTensity of FIRE, Thonicke et al., 2010) is a process-based fire module, used in various vegetation models (e.g. Lasslop et al., 2014; Yue and Unger, 2018), including LPJmL4. We describe here its main features, which are published in Thonicke et al. (2010). SPITFIRE calculates fire disturbance by simulating the ignition, the danger, the spread and the effects of fire separately. As ignition sources SPITFIRE considers human ignition and lightning flashes. Human ignitions ($n_{h,ig}$) are calculated as a function of population density:

$$n_{h,ig} = P_D \cdot k(P_D) \cdot a(N_D)/100, \tag{1}$$

where

$$k(P_D) = 30.0 \cdot \exp(p_h \cdot \sqrt{P_D}). \tag{2}$$

$P_D$ is the human population density (individuals km$^{-2}$) and $a(N_D)$ (ignitions individual$^{-1}$ day$^{-1}$) describes the inclination of humans to cause fire ignitions (Eq. 3 and 4 in Thonicke et al., 2010). $p_h$ is a parameter, which is set to -0.5 in Thonicke et al. (2010). This relationship assumes that human ignitions are lowest on very low populated regions and on high populated regions through a higher level of urbanization and landscape fragmentation. The ignition is highest for a medium-small population density. Lightning-caused ignitions are prescribed by lightning data from the OTD/LIS Gridded Climatological data set

(Christian et al., 2003), assuming that 20 % of the flashes reach the ground and 4 % of cloud-to-ground strikes can start a fire. In the study area of South America human ignitions are by far the most dominant ignition source, due to missing lightning in the dry season.

Fire danger is by default computed by using the Nesterov index which accounts for the maximum and dew point temperatures as well as scaling factors for different PFTs on a daily time step. In the following section, we describe the calculation of the fire danger indices in detail (Sect. 2.2). Fire duration $t_{fire}$ (min) is calculated as a function of the fire danger index, assuming that fires burn longer under a high fire danger:

$$t_{fire} = \frac{241}{1 + 240 \cdot \exp(p_t \cdot FDI)}, \tag{3}$$

where $p_t$ is set to -11.06 in Thonicke et al. (2010). The maximum fire duration per day is 240 minutes.

The calculation of the forward rate of spread $ROS_{f,surface}$ (m min$^{-1}$) is based on the Rothermel equations (Rothermel, 1972; Pyne et al., 1996; Wilson, 1982):

$$ROS_{f,surface} = \frac{I_R \cdot \xi \cdot (1 + \Phi_w)}{\rho_b \cdot \epsilon \cdot Q_{ig}}, \tag{4}$$

where $I_R$ is the reaction intensity, $\xi$ the propagation flux ratio, $\Phi_w$ a multiplier that accounts for the effect of wind, $\epsilon$ the effective heating number, $Q_{ig}$ the heat of pre-ignition and $\rho_b$ the fuel bulk density (Eq. 9 in Thonicke et al., 2010). $\rho_b$ (kg m$^{-3}$) is a PFT-dependent parameter and describes the density of the fuel, which is available for burning. It is weighted over the different fuel classes. Hence, a changing PFT distribution has an impact on $ROS_{f,surface}$.

The simulated fire ignitions, fire danger and fire spread are then used to calculate the burnt area, fire carbon emissions, and plant mortality. Plant mortality depends on the scorch height (*SH*) and the probability of mortality due to crown damage $P_m(CK)$. *SH* describes the height of the flame at which canopy scorching occurs. It increases with the 2/3 power of the surface intensity $I_{surface}$:

$$SH = F \cdot I_{surface}^{0.667}, \tag{5}$$

where *F* is a PFT-dependent parameter. Assuming a cylindrical crown, the proportion *CK* affected by fire is calculated as:

$$CK = \frac{SH - H + CL}{CL}, \tag{6}$$

where *H* is the height of the average woody PFT and *CL* the crown length. The probability of mortality $P_m(CK)$ due to crown damage is then calculated by:

$$P_m(CK) = rCK \cdot CK^p, \tag{7}$$

where *rCK* is a PFT depended resistance factor between 0 and 1, and *p* in the range of 3 to 4. Disturbance by fire mortality has a large impact on the vegetation dynamics, which are calculated within LPJmL. SPITFIRE further includes a surface intensity threshold ($10^6$, fraction burnt area per grid cell), which describes the threshold of the possible area burnt below which the surface intensity is set to zero and hence burnt area, emissions and fuel consumption is set to zero.

SPITFIRE considers anthropogenic effects on fire by taking into account human ignitions but does not account for fire suppression. Only wildfires occurring in natural vegetation are simulated. Fire on managed land like agriculture or pasture areas is not implemented, which has to be taken into account if simulated burnt area is compared with satellite observation.

Furthermore, we introduced a small technical change in the LPJmL4 interaction with SPITFIRE compared to the original
SPITFIRE implementation: In the version 4.0 of LPJmL the fire litter routine calculates the leaf and litter carbon pools in a daily time step. Since the LPJmL tree allocation works at a yearly time step, this implementation leads to an incorrect LPJmL4-SPITFIRE interaction. We now split the fire-litter routine into two parts; the first one allocates burnt matter into the litter at a daily time step without recalculating the pools and the second one calculates the leaf and root carbon pools at a yearly time step.

## 2.2   Fire danger indices

The fire danger index (FDI) is a key parameter within process-based fire models such as SPITFIRE. The FDI determines the probability and the intensity of a spreading fire, which impacts fire behavior.

### 2.2.1   Nesterov index-based fire danger index ($FDI_{NI}$)

The fire danger index within SPITFIRE is based on the daily (d) calculated Nesterov Index *NI(d)* (Venevsky et al., 2002), which
is widely used in numeric fire simulations. The NI is a cumulative function of daily maximum temperature $T_{max}(d)(°C)$ and dew-point temperature $T_{dew}(d)(°C)$ and set to zero at a precipitation $\geq 3$ mm or a temperature $\leq 4\,°C$:

$$NI(d) = \sum T_{max}(d) \cdot (T_{max}(d) - T_{dew}(d)), \tag{8}$$

$$T_{dew} = T_{min}(d) - 4. \tag{9}$$

The resulting fire danger index has been calculated as in Schaphoff et al. (2018a) (slightly different compared to Thonicke et al. (2010)) by taking into account the NI as measure for weather conditions and a PFT-dependent scaling factor $\alpha_{NI_i}$:

$$FDI_{NI} = max\left(0, 1 - \frac{1}{m_e}exp\left(-\frac{\sum \alpha_{NI_i}}{n} \cdot NI\right)\right), \tag{10}$$

where n is the number of PFTs and $m_e$ the moisture of extinction, which is a PFT-dependent parameter and is weighted over the litter amount. We will use the scaling factors $\alpha_{NI_i}$ in the parameter optimization (Sect. 2.4).

### 2.2.2   Vapor pressure deficit-based fire danger index ($FDI_{VPD}$)

We implemented a new fire danger index, based on the water vapor pressure deficit (VPD). The VPD describes the difference of the saturation water vapor pressure $e_s$ and the actual water vapor pressure in the air. For the parameterization of the VPD

we used an approach based on Pechony and Shindell (2009):

$$VPD \propto 10^{Z(T)}(1 - RH/100), \tag{11}$$

where $T$ is the air temperature, $RH$ the relative humidity and $Z$ the Goff-Gratch equation (Goff and Gratch, 1946) to calculate the saturation vapor pressure. The flammability $F$ at time step $t$ for each grid cell can then be expressed as:

$$F(t) = 10^{Z(T(t))} \left(1 - \frac{RH(t)}{100}\right) VD(t) e^{-c_R R(t)}, \tag{12}$$

where $VD$ is the vegetation density, $R$ the total precipitation in mm/day and $c_R$ is a constant factor ($c_R = 2$ day/mm). Here we used the simulated $FPC$ from LPJmL4 as a proxy for the $VD$. The soil is a natural buffer for drought periods and heavy rainfall events. In the Nesterov index this was taken into account by the cumulative nature of this index. Since the VPD-based fire danger index is not cumulative, this buffering effect is taken into account by taking the monthly mean of the precipitation. In doing so we avoid unrealistic high flammability fluctuations in time steps with isolated events of very low or very high precipitation ($R$).

Based on this implementation in SPITFIRE, the resulting FDI was much smaller than the original $FDI_{NI}$. Hence, we scaled the VPD up with a PFT-dependent scaling factor $\alpha_{VPD_i}$, weighted over the corresponding FPC:

$$FDI_{VPD} = \frac{\sum \alpha_{VPD_i} \cdot FPC_i}{\sum FPC_i} \cdot F(t). \tag{13}$$

$\alpha_{VPD_i}$ for the $FDI_{VPD}$ was not included in (Pechony and Shindell, 2009), but is important in order to allow different fire responses for different tree and grass types. We will use the scaling factors $\alpha_{VPD_i}$ in the parameter optimization (Sect. 2.4). In comparison to the NI, the $FDI_{VPD}$ requires more climate variables as input as it uses relative humidity and vegetation cover as additional fire-relevant variables. Vegetation cover has a direct link to fire risk by providing the number of available fuel for burning. According to many studies (e.g. Ray et al., 2005; Sedano and Randerson, 2014; Seager et al., 2015) the $FDI_{VPD}$ is a very accurate fire danger index with a high correlation with fire occurrence, while still being relatively easy to implement in a global fire model.

The general behavior of the two indices as modelled by LPJmL in dependence of relative humidity and temperature is shown in Fig. 3. The Nesterov index shows a strong but very localized maximum for high temperatures and a small humidity. Hence a spreading fire is only possible in a very small climate range (here ca. from 25° Celsius and a relative humidity smaller than 0.5). The VPD on the other hand shows a less pronounced maximum but a medium fire danger also for wetter and colder regions. The slope of towards lower VPD values is also smaller compared to the Nesterov index. Especially in regions with temperatures colder than 20°C and a relative humidity smaller than ca. 0.6 a fire is still possible. This might increase the area in which fires can occur compared to the Nesterov index, which could be an important improvement, enabling SPITFIRE to simulate more fire in wetter and colder regions. The calculated VPD and NI values shown in Fig. 3 are based on a LPJmL-SPITFIRE run, and thus the influence of vegetation distribution on both fire danger indices.

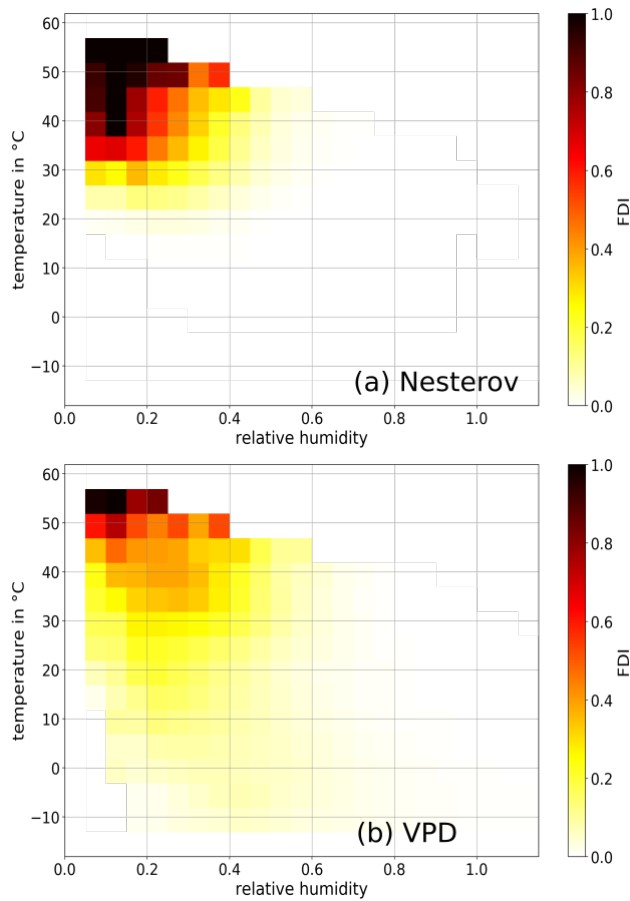

**Figure 3.** Dependence of the simulated fire danger index on monthly mean relative humidity and temperature for (a) the Nesterov-based index and (b) the VPD-based index. Both indices were calculated with monthly data for the years 2000-2010.

## 2.3 Model input data

LPJmL4-SPITFIRE requires input data on daily air temperature, precipitation, long-wave and shortwave downward radiation, wind and specific humidity, which are taken from the NOAH Global Land Assimilation System (GLDAS, Rodell et al., 2004). The data has a spatial resolution of $0.25° \times 0.25°$ and the time step is 3 hours. We regridded and aggregated the data set to the LPJmL resolution of $0.5° \times 0.5°$ and to a daily time step. We used the GLDAS 2.0 for the years 1948-1999 and the version GLDAS 2.1 for the years 2000-2017. GLDAS 2.1 uses multiple satellite- and ground-based observational data as well as advanced land surface modelling and data assimilation techniques. GLDAS 2.0 is forced entirely with the Princeton meteorological forcing data (Civil and Environmental Engineering/Princeton University, 2006). Because LPJmL4 requires at least 30 years of climate data for its spin-up (Sect. 2.1.1), the time span covered by GLDAS 2.1 is too short. To run the model, we used both climate data sets, but used the years 2003-2013 from GLDAS 2.1 for the optimization and 2005-2015 for the evaluation

period.

Furthermore, LPJmL4-SPITFIRE is forced with gridded constant soil texture (Nachtergaele et al., 2009) and annual information on land use from Fader et al. (2010). Atmospheric $CO_2$ concentrations are used from Mauna Loa station (Quéré et al., 2015) and applied globally. The population density is taken from Goldewijk et al. (2011) and the lightning flashes are taken
from the OTD/LIS satellite product (Christian et al., 2003).

## 2.4 Model optimization

To estimate parameters of LPJmL4-SPITFIRE, we aimed to calibrate model results against satellite observations of burnt area (GFED4: Giglio et al., 2013; Werf et al., 2017). However, as fire occurrence and spread impact and depend on vegetation productivity, hence fuel load, we wanted to ensure to not over-fit LPJmL4 against burnt area but to additionally achieve a
realistic vegetation distribution. Therefore, we additionally included a satellite-derived data set on above-ground biomass of trees (AGB, Avitabile et al., 2016) in the optimization. We combined burnt area and AGB with the corresponding model outputs within a joint cost function and applied a genetic optimization algorithm to estimate model parameters (Fig. 2). The implementation of the genetic optimization algorithm (GENetic Optimization Using Derivatives (GENOUD), Mebane and Sekhon, 2011) for LPJmL is described in Forkel et al. (2014). The used cost function is based on the Kling-Gupta efficiency
(KGE), which is the Euclidean distance in a three-dimensional space of model performance measures that account for the bias, ratio of variance and correlation between simulations and the observations. Gupta et al. (2009) showed that the KGE performs in an optimization setup is better than, e.g., the Nash-Sutcliffe efficiency (and hence MSE). We extended the KGE by defining it for multiple data sets d (i.e. burnt area and AGB):

$$Cost = \sqrt{\sum_{d=1}^{N} \left(\frac{\acute{s_d}}{\acute{o_d}} - 1\right)^2 + \left(\frac{\sigma_{s,d}}{\sigma_{o,d}} - 1\right)^2 + (r(s_d, o_d) - 1)^2} \qquad (14)$$

where $\acute{s}$ and $\acute{o}$ are mean values (bias component) over space (i.e. different grid-cells) and time (e.g. months) of simulations $s$ and the observations $o$, respectively. $\sigma_s$ and $\sigma_o$ are variances (variance component) and $r$ is the Pearson correlation coefficient over space and time. The optimization was performed for 40 grid-cells in South America to represent a variety of fire regimes (Fig. 2). We selected the grid cells manually to cover active fire regions (either in the model or in the evaluation data), specifically in the Cerrado and Caatinga. We selected a high density of grid cells in the Caatinga region to improve the very poor
model performance in this region. To make sure that the model performance in the Caatinga and Cerrado was not achieved at the cost of a poor performance in other areas, we also additionally selected some cells in areas where initial fire modeling gave good results, as well as in areas where minimal or no fire occurs (central Brazilian Amazon). After inspection of the results, minor adjustments were made and the selection of the grid cells was modified to account for neglected regions (which showed worsening of the model performance). These initial analysis actually demonstrate that the choice of grid cells is important
for the model optimization and requires the development of a more thorough selection method in future model optimization applications.

Several parameters of LPJmL4-SPITFIRE were included in the optimization that cover different fire processes (see Tab. 2): ignition (human ignition parameter $p_h$, Eq. 2), fire danger (scaling factors FDI ($\alpha_{NI_i}$ and $\alpha_{VPD_i}$), Eq. 10 and 13), fire spread (fire duration $p_t$, Eq. 3), fuel bulk density ($\rho_b$, Eq. 4), surface intensity threshold and fire effects (scorch height parameter F,

Eq. 5; crown mortality parameter rCK, Eq. 7). While $p_t$, $p_h$ and the surface intensity threshold are global parameters (for all PFTs), the others were optimized for each PFT separately. Since we focus here on tropical South America, we used tropical broadleaved evergreen (TrBE), tropical broadleaved raingreen (TrBR) and tropical herbaceous (TrH) PFTs for the optimization.

In genetic optimization algorithms, each model parameter is called an individual with a corresponding fitness, which represents the cost of the model against the observations. At the beginning of the optimization process, the GENOUD algorithm creates a generation of individuals based on random sampling of parameter sets within the prescribed parameter ranges. After the calculation of the cost of all individuals of the first generation, a next generation is generated by cloning the best individuals, by mutating the genes or by crossing different individuals (Mebane and Sekhon, 2011). This results, after some generations, in

a set of individuals with highest fitness, i.e. parameter sets with minimized cost. To find an optimum parameter set also used the BFGS gradient search algorithm (named after the authors Broyden, 1970; Fletcher, 1970; Goldfarb, 1970; Shanno, 1970) within the GENOUD algorithm. An optimized parameter set of the BFGS algorithm is used as individual in the next generation. We were applying the GENOUD algorithm with 20 generations and a population size of 800 individuals per generation, which corresponds to 16000 single model runs. We decided on this amount of iterations, because the cost kept almost constant in the

last iterations and the parameter values did not change to the 6th digit, beyond which changes are not really relevant for model applications. During the optimization we ran the model parallel for each grid cell (40 grid cells and CPU's, 3.2GHz) and had a total optimization time of ca. 24 hours.

The comparison of the two presented fire danger indices is the main objective of this study. Hence the optimization of the PFT-dependent FDI scaling factors $\alpha_{NI_i}$ and $\alpha_{VPD_i}$ is crucial and obligatory for the VPD because of no prior values. Accord-

ingly, we conducted two different optimization experiments using LPJmLmdi: First, using a FDI based on the VPD (VPD$_{optim}$ hearafter) and secondly using the a FDI based on the NI (NI$_{optim}$ hereafter). Both resulting parameter sets were then used for LPJmL4 runs and were compared to the unoptimized original model version using the NI (NI$_{orig}$ hereafter) and various evaluation data sets.

## 2.5 Evaluation data

We used burnt area from the global fire emission database (GFED4; Giglio et al., 2013; Werf et al., 2017), in the model optimization and to evaluate model results. The global data set is available at a resolution of $0.25° \times 0.25°$ in a monthly time step from 1997 until 2016. The GFED burnt area product is based on the 500 m Collection 5.1 MODIS direct broadcast (DB) burnt area product (MCD64A1, after 2001). We used data for the years 2003-2013 in the optimization in order to not include

potential inconsistencies between the GLDAS 2.0 and 2.1 climate data sets or between burnt area observations within GFED that originate from different satellite sensors. The GFED product comes with a stratification of burnt area by land cover from the MODIS land cover map in the resolution of 500 m (Giglio et al., 2013). As LPJmL does not simulate fire on managed lands, we excluded burnt area on cropland classes from model-data comparisons. Due to lack of data we however did not account for the proportion of pastures. To constrain the simulated vegetation distribution, we used the AGB data set from Avitabile et al.

(2016). This data set is approximately representative for the late 2000s and therefore we compared it against simulated AGB for the years 2009-2011. We regridded all data set to a $0.5° \times 0.5°$ resolution. In addition, we used maps of PFTs as derived from the ESA CCI land cover map V2.0.7 (Li et al., 2018; Forkel et al., 2014).

## 2.6 Evaluation metrics

To quantify the performance of the model output, we applied the Pearson Correlation between two time series, the normalized mean square error (NMSE; Kelley et al., 2013) and the Willmott coefficient of agreement (W; Willmott, 1982) to describe differences between the model simulation and the reference data sets. The NMSE is calculated by:

$$NMSE = \frac{\sum_{i=1}^{N}(y_i - x_i)^2}{\sum_{i=1}^{N}(x_i - \bar{x})^2} \tag{15}$$

where $y_i$ is the simulated and $x_i$ the observed value in the grid cell $i$. $\bar{x}$ is the mean observed value. The NMSE is zero for perfect agreement between simulated and modelled results, 1.0 if the model is as good as using the observed mean as a predictor and larger than 1.0 if the model performs worse than that. We chose the NMSE to represent and compare the model errors, as it has a squared error term, which puts a stronger emphasis on large deviations between simulations and observations as compared to a linear term, and due to its normalization it is comparable across different parameters. Especially for fire simulations we have a relatively large deviation between simulations and observations.

The Willmott coefficient of agreement is given by:

$$W = 1 - \frac{\sum_{i=1}^{N}(y_i - x_i)^2 \cdot A_i}{\sum_{i=1}^{N}(|y_i - \bar{x}| + |x_i - \bar{x}|)^2 \cdot A_i} \tag{16}$$

which additionally accounts for the area weight $A_i$ of the grid cell $i$. The Willmott coefficient is a squared index, where a value of 1 stands for perfect agreement between simulated and modelled runs and gets smaller for worse agreements with a minimum of 0. Unlike the coefficient of determination, the Willmott coefficient is additionally sensitive to biases between simulations and observations.

## 3 Results

### 3.1 Performance of optimized fire danger index formulations

Overall, the yearly burnt area simulated by the standard SPITFIRE model (using the original Nesterov index, $NI_{orig}$) showed poor simulation results over South America as compared to the GFED4 evaluation data set (Fig. 4 a and b: NMSE=1.80, W=0.27). The average yearly burnt area (without croplands) for South America was with ca. 14 million ha about 25 % smaller than the observed burnt area with 19 million ha in the shown period from 2005-2015. The spatial pattern of the modelled burnt area agreed well with the GFED4 data in the region of the Cerrado that is close to the Caatinga border, while the fires in other semi-arid regions of the continent were underestimated. For example, simulated fire is underestimated in the savanna-areas in the northern part of South America (on the Columbian-Venezuelian border) even though there is a strong signal visible in

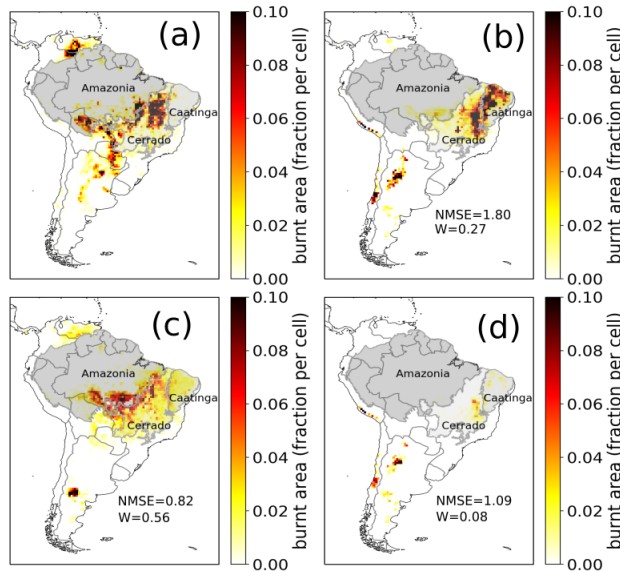

**Figure 4.** Yearly burnt area over a mean from 2005-2015 as fraction per cell. (a) GFED4 evaluation data of burnt area excluding crops and simulated burnt area by SPTIFIRE using the (b) $NI_{orig}$ version, (c) $VPD_{optim}$ version, (d) $NI_{optim}$ version

the satellite observations. The biomes Caatinga and Cerrado, which are of special interest in this study, showed very different results: while fire in Caatinga was overestimated, it was underestimated in the Cerrado.

The optimized version using $NI_{optim}$ (Fig. 4 d), led to an overall decrease of fire, with a slight improvement of NMSE (1.09) as compared to $NI_{orig}$ and a worse Willmott coefficient of 0.08. While the overestimation of fire in Caatinga was reduced, all the

fires across South America also have decreased significantly, which led to a general underestimation of fire by 90 % (2 million ha). The optimized version, using $VPD_{optim}$ (Fig. 4 c), clearly improved the model performance, mainly by shifting much of the simulated burnt area from the sparsely vegetated Caatinga towards the Cerrado region (NMSE=0.82 and W=0.56). In addition, by using $VPD_{optim}$, the model results also showed fire occurrence in northern South America, where fire was not at all or only minimally simulated when using $NI_{optim}$ or $NI_{orig}$. The total burnt area was in this model version ca 20 % smaller

than the evaluation data set (16 million ha).

The burnt area time series from 2005 to 2015 provides a more detailed view on the model performance for the fire-prone Cerrado and Caatinga region (Fig. 5). While model performance was relatively good for the Cerrado region with $NI_{orig}$ (NMSE=0.3, W=0.89, $R^2$=0.78), the simulated burnt area was strongly overestimated in the Caatinga region throughout the whole period (NMSE=327.82, W=0.14, $R^2$=0.59). After the optimization of the NI, the model performance indeed improved

for the Caatinga (NMSE=1.07, W=0.73, $R^2$=0.31), but at the same time the performance for the Cerrado declined (NMSE=1.07, W=0.36, $R^2$=0.4). On the other hand $VPD_{optim}$ showed an improved fire representation compared to the standard settings in the Cerrado (NMSE=0.27, W=0.9, $R^2$=0.8) as well as in the Caatinga (NMSE=15.2, W=0.46, $R^2$=0.56). Even though fire in the Caatinga was still overestimated, the NMSE decreased by a factor of six.

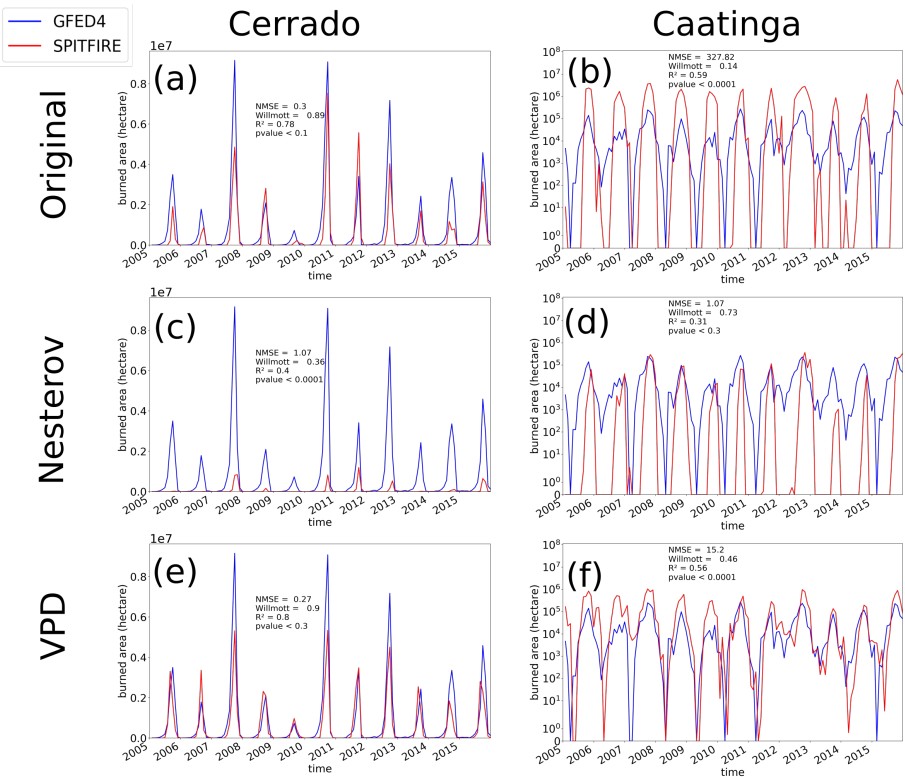

**Figure 5.** Time-series of monthly burnt area from 2005 - 2015 simulated by SPITFIRE (red lines) compared to GFED4 evaluation data (blue lines) for: (a) The Cerrado region, using $NI_{orig}$. (b) The Caatinga region, using the $NI_{orig}$. (c) The Cerrado region, using $NI_{optim}$. (d) The Caatinga region, using $NI_{optim}$. (e) The Cerrado region, using $VPD_{optim}$. (f) The Caatinga region, using $VPD_{optim}$. Note the logarithmic scale for the Caatinga, which was applied in order to account for the large differences between the different model versions (for a non-logarithmic version see Fig. A6).

Overall, the total amount of burnt area in the Cerrado was for all three model versions smaller than in the evaluation data set. Fire occurrence in the Caatinga was, on the other hand, largely overestimated by the $NI_{orig}$ and the $VPD_{optim}$ version. Just in the $NI_{optim}$ version the burnt area of the Caatinga is in the same order of magnitude as the evaluation data set, which also led, however, to a large underestimation in the Cerrado (Tab. 1 and Fig. 4). Also the Amazonia region mostly improved
5 by using the $VPD_{optim}$ version (Tab. 1, Fig. A3). The $R^2$ and the Willmott coefficient improved, while the NMSE increased slightly. With the Nesterov index fire was strongly underestimated in the Amazon region, while the optimized VPD fixes this underestimation. The fire is only modelled (and also observed, see Fig. 4) at the edges to the Amazon, where wood density is lower and deforestation takes place. In the closed continuous forest area towards the center of the Amazon almost no fire is observed and neither modelled. The total burnt area increased from 0.7 million ha to 4.8 million ha (for $VPD_{optim}$) , which is
10 now a bit overestimated to the observed burnt area of 3.4 million ha. Using the $NI_{optim}$ all error metrics as well as the total burnt area decreased.

**Table 1.** Comparison of the burnt area results in terms of NMSE, the Willmott coefficient of agreement and the sum (in ha per year) between $NI_{orig}$, $VPD_{optim}$, $NI_{optim}$ and the GFED evaluation data

| Region | NMSE | Willmott | Sum |
|---|---|---|---|
| **Spatial - South America** | | | |
| GFED | | | $1.9 \cdot 10^7$ |
| $NI_{orig}$ | 1.80 | 0.27 | $1.4 \cdot 10^7$ |
| $VPD_{optim}$ | 0.82 | 0.56 | $1.6 \cdot 10^7$ |
| $NI_{optim}$ | 1.09 | 0.08 | $0.2 \cdot 10^7$ |
| **Temporal - Cerrado** | | | |
| GFED | | | $9.2 \cdot 10^6$ |
| $NI_{orig}$ | 0.30 | 0.89 | $5.2 \cdot 10^6$ |
| $VPD_{optim}$ | 0.27 | 0.90 | $6.4 \cdot 10^6$ |
| $NI_{optim}$ | 1.07 | 0.36 | $0.6 \cdot 10^6$ |
| **Temporal - Caatinga** | | | |
| GFED | | | $0.4 \cdot 10^6$ |
| $NI_{orig}$ | 327.82 | 0.14 | $6.0 \cdot 10^6$ |
| $VPD_{optim}$ | 15.2 | 0.46 | $1.6 \cdot 10^6$ |
| $NI_{optim}$ | 1.07 | 0.73 | $0.3 \cdot 10^6$ |
| **Temporal - Amazonia** | | | |
| GFED | | | $3.4 \cdot 10^6$ |
| $NI_{orig}$ | 0.83 | 0.56 | $0.7 \cdot 10^6$ |
| $VPD_{optim}$ | 0.93 | 0.83 | $4.8 \cdot 10^6$ |
| $NI_{optim}$ | 1.22 | 0.32 | $0.02 \cdot 10^6$ |

## 3.2 Optimized model parameters

Seven fire-related parameters were optimized, in order to improve the fire representation in the LPJml4-SPITIFRE model. Here we compare the optimized parameters for the different model versions in order to evaluate and discuss parameter variability and changes. Table 2 shows all parameters, used for the optimization, their lower and upper boundary and the resulting optimized value. Since the FDI directly controls the amount of modelled fire, the FDI scaling factors for the different PFTs are central for this analysis. For both optimization experiments the boundaries were, hence, set rather generously within one magnitude of the original value. In the $NI_{optim}$ experiment, all scaling factors generally decreased compared to the standard values used for $NI_{orig}$. Here, TrH displayed the smallest scaling factor ($9.39 \cdot 10^{-6}$), followed by TrBE ($2.48 \cdot 10^{-5}$) and TrBR ($4.76 \cdot 10^{-5}$). Since the VPD is a newly implemented fire danger index, we have no standard values to compare the optimized scaling factors with. Here, the TrBE showed the largest value (22.41), ca. 20 times as large as the TrBR (1.21) and TrH (1.13) (Tab. 2).

**Table 2.** All optimized parameters with their standard values, the upper and lower boundary of the parameter ranges and the resulting optimized value including parameter for specific PFTs and global parameter, which have the same value for all PFTs. All parameters exept $\rho_b$ have no unit.

| Parameter | PFT | Standard value (as in Thonicke et al., 2010) | Lower bound. | Upper bound. | After optimization |
|---|---|---|---|---|---|
| **NI$_{optim}$** | | | | | |
| scaling factor FDI $\alpha_{NI_i}$ | TrBE | $3.34 \cdot 10^{-5}$ | $7 \cdot 10^{-6}$ | $1.33 \cdot 10^{-4}$ | $2.4885 \cdot 10^{-5}$ |
| scaling factor FDI $\alpha_{NI_i}$ | TrBR | $3.34 \cdot 10^{-5}$ | $7 \cdot 10^{-6}$ | $1.33 \cdot 10^{-4}$ | $4.7649 \cdot 10^{-5}$ |
| scaling factor FDI $\alpha_{NI_i}$ | TrH | $6.67 \cdot 10^{-5}$ | $7 \cdot 10^{-6}$ | $1.33 \cdot 10^{-4}$ | $9.3949 \cdot 10^{-6}$ |
| fire duration parameter $p_t$ | all PFTs | -11.06 | -13 | -9 | -9.0011 |
| scorch height parameter F | TrBE | 0.1487 | 0.01 | 0.6 | 0.1282 |
| scorch height parameter F | TrBR | 0.061 | 0.01 | 0.6 | 0.0752 |
| crown mortality parameter rCK | TrBE | 1.0 | 0.5 | 1 | 0.5030 |
| crown mortality parameter rCK | TrBR | 0.05 | 0 | 0.5 | 0.4038 |
| fuel bulk density $\rho_b$ (kg m$^{-3}$) | TrBE | 25.0 | 22.5 | 27.5 | 26.6473 |
| fuel bulk density $\rho_b$ (kg m$^{-3}$) | TrBR | 13.0 | 11.7 | 14.3 | 13.1896 |
| fuel bulk density $\rho_b$ (kg m$^{-3}$) | TrH | 2.0 | 1.8 | 2.2 | 2.0019 |
| human ignition parameter $p_h$ | all PFTs | -0.5 | -0.6 | -0.4 | -0.5426 |
| surface intensity threshold | all PFTs | $10^{-6}$ | $10^{-7}$ | $10^{-5}$ | $1.0317 \cdot 10^{-6}$ |
| **VPD$_{optim}$** | | | | | |
| scaling factor FDI $\alpha_{VPD_i}$ | TrBE | - | 1 | 50 | 22.4181 |
| scaling factor FDI $\alpha_{VPD_i}$ | TrBR | - | 1 | 50 | 1.2135 |
| scaling factor FDI $\alpha_{VPD_i}$ | TrH | - | 1 | 50 | 1.1299 |
| fire duration parameter $p_t$ | all PFTs | -11.06 | -13 | -9 | -11.3753 |
| scorch height parameter F | TrBE | 0.1487 | 0.01 | 0.6 | 0.1930 |
| scorch height parameter F | TrBR | 0.061 | 0.01 | 0.6 | 0.0799 |
| crown mortality parameter rCK | TrBE | 1.0 | 0.5 | 1 | 0.9983 |
| crown mortality parameter rCK | TrBR | 0.05 | 0 | 0.5 | 0.4801 |
| fuel bulk density $\rho_b$ (kg m$^{-3}$) | TrBE | 25.0 | 22.5 | 27.5 | 22.5923 |
| fuel bulk density $\rho_b$ (kg m$^{-3}$) | TrBR | 13.0 | 11.7 | 14.3 | 13.3750 |
| fuel bulk density $\rho_b$ (kg m$^{-3}$) | TrH | 2.0 | 1.8 | 2.2 | 1.8944 |
| human ignition parameter $p_h$ | all PFTs | -0.5 | -0.6 | -0.4 | -0.5332 |
| surface intensity threshold | all PFTs | $10^{-6}$ | $10^{-7}$ | $10^{-5}$ | $3.6317 \cdot 10^{-6}$ |

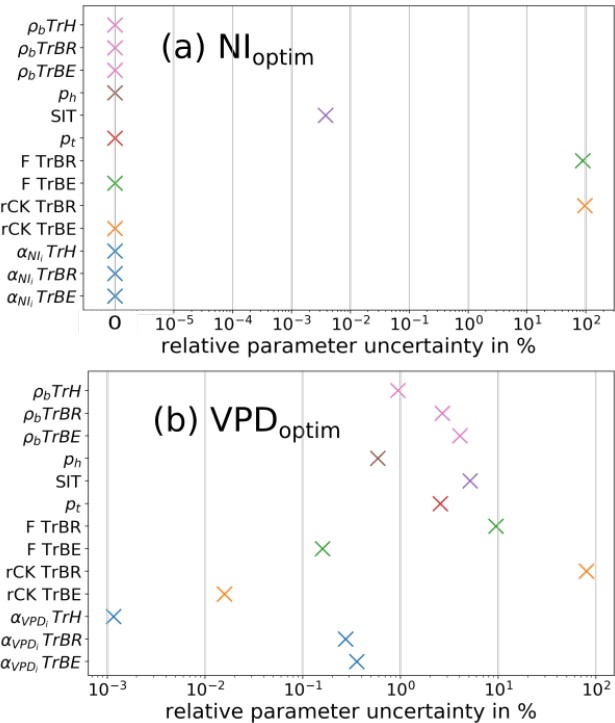

**Figure 6.** Relative uncertainty of model parameters after optimization for (a) $NI_{optim}$ and (b) $VPD_{optim}$. The relative uncertainty is the ratio of the uncertainty after the optimization (range of all parameter sets with low cost, below the 0.05 quantile) divided by the uncertainty before the optimization (range of the parameters for the optimization). Low and high values of relative uncertainty indicate strongly and weakly constrained parameters, respectively. SIT denotes the surface intensity threshold. PFT-dependent parameters are grouped with the same color.

In case of the other optimized parameters the boundaries were set smaller in order to decrease the possibility that a large error in the estimation of several parameters would lead to a better overall cost in the optimization procedure. The human ignition parameter became smaller for both optimizations, which led to a smaller amount of human ignitions (from -0.5 to -0.54 in $NI_{optim}$ and -0.53 in $VPD_{optim}$). The fuel bulk density increased for all three tropical PFTs in the $NI_{optim}$ version,

5     while for $VPD_{optim}$ the fuel bulk density of the TrBE and TrH PFTs decreased and for the TrBR increased. For the $NI_{optim}$ version, the fire duration parameter ($p_t$) increased, leading to a shorter fire duration (from -11.06 to -9), while the value for the $VPD_{optim}$ version stayed relatively similar (-11.37) to the prior value. The surface intensity threshold became slightly larger for the $NI_{optim}$ version than the original value (from $10^{-6}$ to $1.03 \cdot 10^{-6}$). For $VPD_{optim}$ the parameter increased by a factor of three ($3.63 \cdot 10^{-6}$). The mortality related parameters $F$ and $rCK$ led in the $NI_{optim}$ version both to a decrease in the fire-related

10     mortality for TrBE and an increase for TrBR PFTs. The optimized parameters for $VPD_{optim}$ led to a decrease in the fire-related mortality for both PFTs except for the TrBR $rCK$, which led to an increased mortality.

The relative uncertainties were for most optimized parameters very small (between 0 and 10%), hence these parameters were strongly constrained (Fig. 6). Just the fire-mortality related parameters ($F$ and $rCK$) had large uncertainties for the TrBR, hence

**Table 3.** Comparison of the results for AGB and the TrBE PFT cover in terms of NMSE and the Willmott coefficient of agreement between $NI_{orig}$, $VPD_{optim}$ and $NI_{optim}$ in South America (SA), in the Cerrado and in the Caatinga.

| Region | NMSE | Willmott |
|---|---|---|
| **AGB** | | |
| SA ($NI_{orig}$) | 0.97 | 0.83 |
| SA ($VPD_{optim}$) | 0.91 | 0.84 |
| SA ($NI_{optim}$) | 0.99 | 0.83 |
| Cerrado ($NI_{orig}$) | 15.06 | 0.25 |
| Cerrado ($VPD_{optim}$) | 12.36 | 0.28 |
| Cerrado ($NI_{optim}$) | 16.06 | 0.24 |
| Caatinga ($NI_{orig}$) | 11.93 | 0.32 |
| Caatinga ($VPD_{optim}$) | 8.57 | 0.36 |
| Caatinga ($NI_{optim}$) | 10.44 | 0.33 |
| **FPC - Evergreen (TrBE)** | | |
| SA ($NI_{orig}$) | 0.42 | 0.82 |
| SA ($VPD_{optim}$) | 0.41 | 0.82 |
| SA ($NI_{optim}$) | 0.43 | 0.81 |
| Cerrado ($NI_{orig}$) | 1.04 | 0.60 |
| Cerrado ($VPD_{optim}$) | 0.70 | 0.64 |
| Cerrado ($NI_{optim}$) | 1.40 | 0.55 |
| Caatinga ($NI_{orig}$) | 1.73 | 0.40 |
| Caatinga ($VPD_{optim}$) | 1.54 | 0.29 |
| Caatinga ($NI_{optim}$) | 2.05 | 0.44 |

were weakly constrained. For $VPD_{optim}$ the uncertainty of *rCK* (TrBR) was 0.8 and for $NI_{optim}$ the uncertainty of *F* was 0.9 and for *rCK* 1 (TrBR).

The decrease in the model error (cost) due to the optimization process has been mainly due to improvement in the burnt area. While for the $NI_{optim}$ the cost of the burnt area dataset improved by 81%, the cost of the biomass dataset improved just by 6%. In case of the $VPD_{optim}$ the cost of the burnt area dataset improved by 49%, whereas the biomass dataset improved by 19% (Fig. A5).

### 3.3 Model evaluation for South America

The modelled above-ground biomass (AGB) of trees in South America was throughout all model versions larger than the evaluation data set indicates (Fig. 7). Especially the biomass in the Amazon region is with an average of ca. 20 kgC/m$^2$ about one third overestimated. The drier savanna regions on the continent yielded a biomass of ca. 5-10 kgC/m$^2$, which also

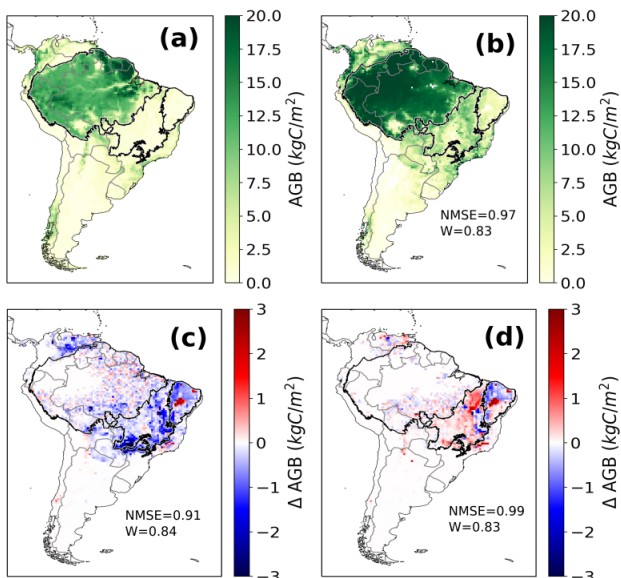

**Figure 7.** Annual above ground biomass (AGB) of trees over a mean from 2005-2015 in kgC/m$^2$. (a) Avitabile evaluation data. (b) Simulated AGB by LPJmL4-SPITFIRE in the NI$_{orig}$ version. (c) Difference between VPD$_{optim}$ and NI$_{orig}$. (d) Difference between NI$_{optim}$ and NI$_{orig}$. Red (blue) color indicates a larger (smaller) biomass after the optimization.

constitutes an overestimation in wide parts of the Cerrado and the Caatinga biome (evaluation data shows between 1-5 kgC/m$^2$, also see Roitman et al., 2018).

The differences among the different model versions are marginal: The VPD$_{optim}$ version had the best performance compared to the evaluation data set (NMSE= 0.91, W=0.84), the NI$_{orig}$ version had the second best performance (NMSE=0.97, W=0.84)

and the NI$_{optim}$ the worst performance (NMSE=0.99, W=0.83). The model optimization scheme focuses on fire parameters, hence the model performance for AGB can only improve in areas, where the fire occurrence has been modelled poorly and the vegetation-fire interactions have improved due to the optimization process. For example in the center of the Amazon rainforest almost no fire is found in the evaluation data nor is simulated. Hence no improvement of burnt area as well as AGB can be achieved. On the other hand, in regions where the modelling error of burnt area is now reduced, this can also improve simulated

AGB, hence vegetation-fire interactions. In the fire-prone Caatinga and Cerrado the VPD$_{optim}$ version mostly decreased the biomass by up to 3 kgC/m$^2$, showing a better performance compared to the evaluation data set (e.g. in the Cerrado the NMSE decreased from from 15.06 to 12.36 in the VPD$_{optim}$ version compared to NI$_{orig}$, see Tab. 3).

The modelled foliage projective cover (FPC) showed for all three model versions a strong underestimation compared to the evaluation data set of the TrBE throughout the whole Amazonian region (ca. 50% compared to ca. 100% in the evaluation

dataset). In the fire-prone biomes Cerrado and Caatinga, however, the TrBE PFT was sometimes overestimated (TrBE cover between 0 and 40 %, Fig. 8). In the regions with less TrBE the dominant PFT was mostly TrBR (Cerrado) or TrH (Caatinga) (see Fig. A1 and A2).

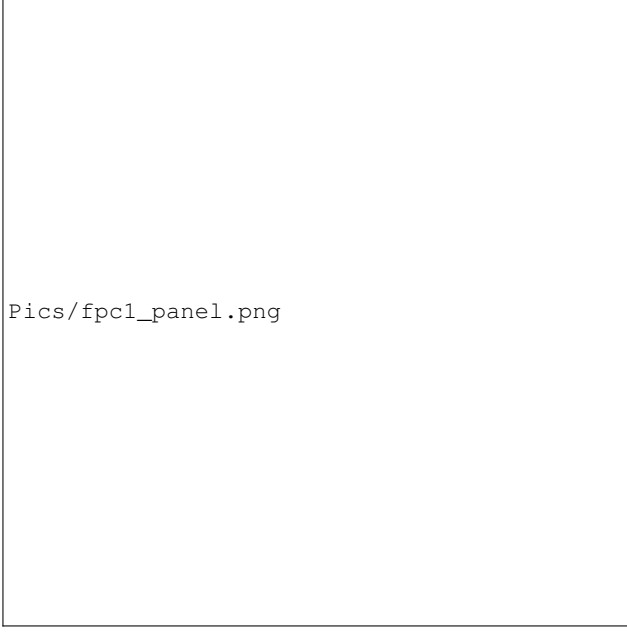

**Figure 8.** Annual FPC cover by tropical broadleaved evergreen PFT over a mean from 2005-2015 as fraction per cell. (a) ESA-CCI evaluation data (b) Simulated FPC by LPJmL4-SPITFIRE using the NI$_{orig}$ version (c) Simulated FPC by LPJmL4-SPITFIRE using the VPD$_{optim}$ version (d) Simulated FPC by LPJmL4-SPITFIRE using the NI$_{optim}$ version

NI$_{optim}$ led to an overall decrease in the model performance also in terms of the TrBE distribution, as both, the NMSE and the Willmott coefficient declined compared to NI$_{orig}$ (NI$_{orig}$: NMSE=0.42, W=0.82; NI$_{optim}$: NMSE=0.43, W=0.81).

The VPD$_{optim}$ version, on the other hand, showed an slightly improved TrBE distribution (NMSE=0.41, W=0.82) but also in this case we obtained an even larger improvement, when only the fire-prone regions Cerrado or Caatinga are considered

5   (Tab. 3). Also for the TrBR and TrH PFT distributions the optimization lead to an improved performance using the VPD$_{optim}$ in the Caatinga and Cerrado, whereas the PFT distribution in the Amazon remained similar to the prior PFT distribution. In the NI$_{optim}$ version, parameter optimization only slightly reduced TrBR cover showing a worse performance compared to VPD$_{optim}$. However, herbaceous cover changed only slightly in all optimization experiments (Fig. A1 and A2).

## 4   Discussion

10   In summary, our results show that the implementation of a new fire danger index based on the water vapor pressure deficit $FDI_{VPD}$ and its optimization against satellite data sets improved the simulations of fire in LPJmL4-SPITFIRE, both in terms of spatial patterns as well as temporal dynamics of burnt area. In the following, we discuss the model improvements, limitations and recommendations for future improvements of process-based global fire models within the DGVM framework.

## 4.1 Improvements in model performance

The VPD results showed a better model performance for fire in the spatial dimension, as well as in the temporal dimension (Tab. 1 and 3). Compared to the Nesterov index, $FDI_{VPD}$ uses additional climate input as relative humidity and precipitation. In the calculation of the Nesterov index precipitation is just used as a threshold. This leads to a better accounting of the very different climatic conditions among various biomes. Furthermore, the $FDI_{VPD}$ includes a direct representation of the vegetation density. The significance of this has been recently shown by findings of Forkel et al. (2019a) who have emphasized the importance of past plant productivity and fuel production for burnt area. This is particularly important for differentiating between fires in biomes with similar PFT distribution. For example, the vegetation density is much larger in the Cerrado, even though the Caatinga and Cerrado have a similar modelled PFT composition, which provides more fuel and therefore leads to a higher fire danger.

While the seasonal and interannual variability in the Caatinga has improved largely using the $FDI_{VPD}$ (NMSE decreased by a factor of ca. 20), the improvement in the Cerrado was relatively small (NMSE decreased by ca. 10%). This is due to the fact, that the optimization tries to obtain a compromise between the different optimized cells. As the model performance was originally much better for the Cerrado, the largest improvement could be achieved for the Caatinga. We have also chosen a large amount of cells in the Caatinga, because the model performance was here particularly bad. This leads to a large improvement in the time series of the Caatinga region, while the improvement for the Cerrado was less significant. With the Nesterov index fire was strongly underestimated in the Amazonia region, while the optimized VPD increases the modelled burnt area. The fire is only present at the edges of the Amazon (both in model and observation, see Fig. 4), where tree density is lower and deforestation takes place. In the closed continuous forest area towards the center of the Amazon almost no fire is observed and also not simulated.

Another result of the optimizing procedure, using $FDI_{VPD}$, was the improvement of the PFT distribution and the aboveground biomass of trees especially in the fire-prone biomes Caatinga and Cerrado (Fig. 8). For example, the central Amazon, where fire is a scarce event, shows almost no changes compared to the non-optimized model version. Here, it is the improvement of the vegetation model itself, and not the fire module, which can help to improve the model performance of LPJmL4-SPITFIRE. Hence, it emphasizes that we need to include further parameters in the optimization, which impact directly the PFT distribution, biomass and fire to obtain a significant improvement in the spatial and temporal distribution of both, vegetation and fire. However, this study focused solely on the parameters within the SPITFIRE module. Due to the focus on fire related parameter, the cost of the burnt area dataset decreases much more than the cost of the biomass dataset (Fig. A5). Hence we only get a substantial improvement in model performance in semi-arid, fire-prone biomes, where vegetation dynamics and fire are strongly coupled. During the optimization-process most of the optimized parameters were well constrained, except for the mortality-related parameters for the TrBR PFT (Fig. 6). The TrBR PFT is dominant in the fire-prone regions, where the mortality-related parameters have a large impact on vegetation dynamics. Hence, they impact multiple LPJmL routines, which are responsible for the PFT distribution and carbon cycling. This leads in turn to a less certain parameter estimation. In order to better constrain these parameters also the optimization of vegetation model parameters would be necessary to decrease the

uncertainties.

The fire danger index scaling factors ($\alpha_{NI_i}$ and $\alpha_{VPD_i}$) convert the quantified fire risk (NI or VPD) into the actual fire danger (FDI). Both scaling factors thus set the magnitude of the fire danger for the different PFTs. Hence they impact directly the fire spread, burnt area and the number of fires as well as indirectly fire mortality. These very important parameters vary significantly for the different PFTs. TrH has the smallest scaling factor in case of both FDIs, which leads to a lower fire danger compared to the other PFTs. This indicates a prior overestimation of the fire danger of grass in tropical South America, as grasslands are generally parametrized to have a low fire resistance and moisture content and can hence burn very easily. This overestimation, compared to tree PFTs has been decreased by the optimization. In case of the VPD also the TrBR is scaled by a much smaller factor than the TrBE, which leads to a lower fire danger index. This is due to the fact, that the TrBR is dominant in dry and fire-prone regions, which experience frequent fires. Here the burnt area was often overestimated by SPITFIRE (e.g. Caatinga or eastern Cerrado) and is now decreased. On the other hand, a larger FDI for the TrBE allows more fire in wetter regions at the edge between the Cerrado and the Amazon rainforests, where TrBE is more dominant. The mortality risk of TrBE for $VPD_{optim}$ remains close to the prior value of 1, confirming previous assumptions about its fire sensitivity. Whereas the rCK for TrBR increased to 0.48, close to the upper boundary, meaning that a mortality risk of 50% when the full crown is scorched and a 7% mortality risk when 50% of the crown is scorched, which makes the TrBR less resistant against crown damage than before. Due to this changes the overestimation of biomass in the original model for the Cerrado/Caatinga region decreased (see Fig. 7).

## 4.2 Limitations of the optimization process

Generally, optimizing a model against burned area is challenging because 1) of the skewed statistical distribution of burned area and 2) because temporal or spatial mismatches in simulated burning can cause large model-data errors. These issues can be avoided with the choice of an appropriate cost function. For example, squared-error metrics tend to underestimate the variance of burned area in comparison to, e.g., the Kling-Gupta efficiency as it has been shown in the optimization of an empirical model for burned area (see Tab. A3 in Forkel et al. (2017)). Here, the optimum parameter set for the Nesterov index-based model resulted in almost no fires across South America. Thereby the optimization algorithm tries to decrease the model error by tending towards a conservative 'no fire strategy' for all biomes. This result nicely demonstrates the need to evaluate model optimization results against spatially and temporally independent data and independent variables (Keenan et al., 2011).

The Nesterov index is not able to capture fire variability within the Caatinga as well as the Cerrado at the same time. This shows that the difference in the PFT distribution between these two biomes is not adequately modelled by LPJmL or just using PFT dependent scaling factors did not sufficiently improve the model performance when using the Nesterov index. On the other hand, using the VPD fire danger index reduced the model error for burned area in both biomes, by improving the modelled performance for the Caatinga and maintaining the good performance of the Cerrado region. Since improved performance of the fire model mainly had minor effect on improving FPC of the tropical PFTs, the presented optimization scheme has to go along with process-based improvements in both, in the fire and in the vegetation modules of LPJmL.

Fire largely depends on the vegetation type and their associated flammability, fire tolerance and mortality. Hence an accurately

modelled vegetation distribution is crucial for a good model performance in terms of burnt area and fire effects (Forkel et al., 2019a; Rogers et al., 2015). As shown in Fig. 8, A1 and A2, the modelled PFT coverage showed an equal distribution of tropical raingreen and evergreen PFTs throughout wide parts of central-northern South America. Evaluation data shows, however, an TrBE dominance in the wet rainforest regions and a TrBR dominance in the Cerrado and Caatinga. This emphasizes the

potential to improve the fire modelling further, based on an improved PFT distribution. In the tropical rainforest the TrBR proportion is overestimated, which leads to problems in the optimization procedure, since TrBR has very different effects on fire spread and is more fire-tolerant (different fuel characteristics and resulting fire intensity). This leads to a lower fire-related mortality, which fits better to the drier and fire prone savanna-like regions (e.g. Cerrado). The poorly modelled PFT distribution also is responsible for the overestimation of the burnt area in the Amazon region. Because of the too large fraction of TrBE

in the Cerrado/Caatinga region the scaling factor for this PFT is relatively high. This leads in turn to an overestimation in the Amazon region, where the fraction of the TrBE is larger.

Since the offset is very small, the years 2000-2003 (first three years of GLDAS 2.1, before the optimization period) are enough for the model to recover from the offset and the carbon pools to return to equilibrium. To exclude the possibility that long-term trends within GLDAS 2.0 changed the modelled vegetation state significantly, we tested our optimization also just based on

GLDAS 2.0 data (until 2010) and on GLDAS 2.1 data (2000-2017) only, using the same years for model spinup, optimization and evaluation. Both versions yielded similar results compared to the optimization presented in this study (results not shown). Due to the fact that evaluation data are only available for the last 10-20 years, we are constrained to optimize the model in this relatively short time period. In South America these years were subject to an unusual high amount of severe droughts and other extreme events (Panisset et al., 2017). As a result, an optimization in this period could lead to a worse model performance in

a period with less pronounced droughts. This is due to the non-linear relationship between the drought signal in the input data set and the resulting modelled biosphere behavior. Nonetheless, we were able to improve the interannual variability and hence, the model performance to a great extent for the Caatinga and slightly for the Cerrado and Amazon regions (Fig. 5 and A3). The Cerrado already had a very good modelling performance before the optimization process, which now only slightly improved. The performance of the interannual and seasonal variability of burnt area for total South America improved substantially (Fig.

A3). The optimized SPITFIRE is now better able to simulate accurately the climate dependent seasonal and interannual variability as well as the spatial extent of fire on natural land throughout the fire-prone woodlands of South America.

Systematic optimizations within a model-data integration setup of fire models which are embedded in a DGVM are still very rare. Previously, Rabin et al. (2018) optimized the fire model FINAL.1 within the land-surface model LM3. Our study differs from Rabin et al. (2018) in the conceptual design of the vegetation-fire models and the optimization process. While LM3 has

been run on a 2° longitude by 2.5° latitude, which is much coarser than LPJmL with 0.5° by 0.5°. This difference allows us to account for a locally better climate input, vegetation and fire interaction.While FINAL.1 is a process-based model, many calculations (e.g. the fire spread routine) are done by multiplying the important factors and fitting the resulting values to observational data. SPITFIRE tries to model the important fire variables by simulating the underlying processes, and by taking the influence of climate and the different fire ignitions into account. An advantage of FINAL.1 is the inclusion of agricultural fires

based on a statistical approach. Whereas Rabin et al. (2018) used a local search algorithm (Levenberg-Marquardt algorithm) to

optimize their model, we used a global search algorithm (genetic optimization). Local search algorithms depend on the chosen initial parameter sets and might eventually end up in a local optimum. A genetic optimization algorithm allows to explore the full parameter space and hence gives a higher chance to find the global optimum. However, local search algorithms require less iterations than global search algorithms (300 in Rabin et al. (2018) vs. 16000 in our study). Forkel et al. (2014) tested the

optimization of LPJmL with different optimization algorithms and found that it was not feasible to optimize LPJmL with a local search algorithm. Rabin et al. (2018) ran the model during the optimization process only for the period of 1991-2009, whereas we made complete model runs including 5000 years of spinup in order to get a model equilibrium for each tested parameter combination.

## 4.3 Limitations of fire modelling in LPJmL4-SPITFIRE

In fire-prone regions the interactions between fire and vegetation dynamics are strong, hence are posing a challenge for global fire models embedded in DGVMs. By just focussing on fire-related parameters, an optimization approach can only to a certain extent improve PFT distribution and simulated biomass. For a good fire representation e.g. in the Cerrado and Caatinga, a shrub PFT could further improve the model performance. Most fires in this region occur, where shrub PFTs are abundant. LPJmL tries to account for this by establishing rather small raingreen PFTs as a shrub replacement. A much better option would be a

separate shrub PFT with parameters leading to a high flammability, but also a low fire mortality. An optimization of LPJmL4-SPITFIRE, including shrub PFTs could yield better results than shown in this study.

Fire models embedded in DGVMs should build on a FDI which is complex enough to account for various fire dynamics, while it's parameterization should be simple enough to be accurately applied on a global scale. While the VPD is more complex and takes into account more climatic input as the Nesterov index, it is still relatively easy to implement in a global fire model.

There are various other fire danger indices used for modelling purposes, as well as real fire danger assessment and fire forecast purposes. For example, fire-prone countries have developed their own fire danger indices (e.g. Canada, Australia), which are suited to the unique local fire regimes and vegetation dynamics. In a global modelling approach, however, we need to find one fire danger index, which suits best for all regions of the world and has a relatively easy implementation to decrease computational cost and the number of input data sets (which might be unavailable or uncertain).

Currently, SPITFIRE does not account for fire in managed land like cropland or managed grassland. We accounted for this by excluding cropland fires from the evaluation burnt area data set. We do, however, not account for the proportion of grassland, which is used for e.g. cattle ranching. Since in SPITFIRE fire is not enabled on pasture, our results show a slightly smaller burnt area throughout South America than could be expected with managed land included and hence also compared to the GFED4 evaluation data set. This effect is however small, because pasture lands cover a substantial fraction only in very few grid cells

(e.g. southern Cerrado; Parente et al., 2017). Fire on managed land is generally difficult to predict in a DGVM because the reason and timing of using fire depends less on climatic factors but mostly on social and political decisions which can vary between countries, regions and localities. We expect further improvement of model performance especially in regions of large land-use areas with fires on pastures included (e.g. Rabin et al., 2018; Pfeiffer et al., 2013).

## 5 Conclusions

We significantly improved the fire representation within LPJmL4-SPITFIRE, applied for South America, by implementing a new fire danger index and applying a model-data integration setup to optimize fire-related parameters. We improved the seasonal and interannual variability, as well as the spatial pattern of burnt area in South America. In addition, modelling of related vegetation variables, e.g., the biomass and the PFT distribution in the fire-prone Cerrado and Caatinga biomes have also been improved.

Optimizing fire parameters has its limits due to error propagation of the PFT distribution and hence their fire traits influencing simulated fire spread and behavior. Furthermore, it remains a challenge to find a fire danger index that is physically interpretable and can be applied globally. In this study, the parameter-optimization by using $FDI_{NI}$ led to a large underestimation of fire and a generally worse model performance, when focusing on the Cerrado and Caatinga biome. However, implementing the more complex $FDI_{VPD}$ and optimizing it thereafter, led to an improved model performance compared to the original SPITFIRE implementation for South America. Our results demonstrate that the improvement of model processes, as well as a systematic model-data optimization are required in order to obtain a more accurate fire representation within complex DGVMs, where observations or experimental evidence to constraint fire parameter are scarce. This work highlights the potential for future model-data integration approaches to obtain a better fire model performance in a global setting, based on improved vegetation dynamics within LPJmL4.

*Code availability.* The model code of LPJmL4 is publicly available through PIK's gitlab server at https://gitlab.pik-potsdam.de/lpjml/LPJmL, and an exact version of the code described here is archived under https://doi.org/10.5281/zenodo.3497213. The R-package for LPJmL is publicly availabe at https://gitlab.pik-potsdam.de/lpjml/LPJmLmdi and the exact version of the package used here is archived under https://doi.org/10.5281/zenodo.3497201.

# Appendix A

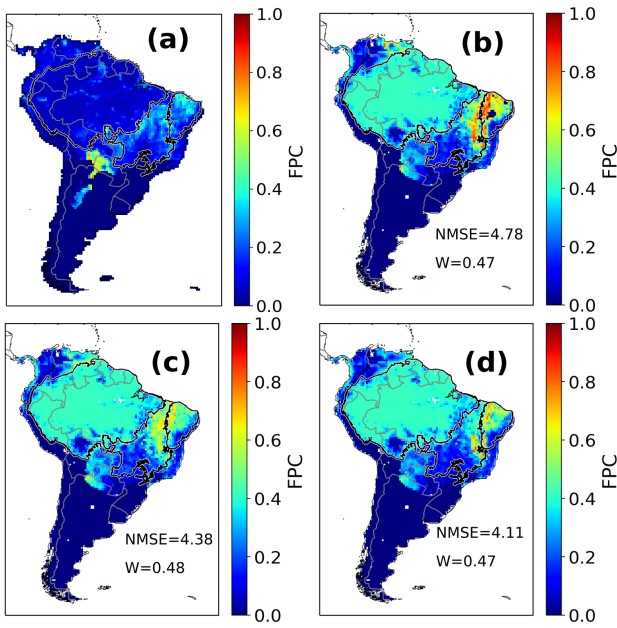

**Figure A1.** Annual FPC cover by tropical broadleaved raingreen PFT over a mean from 2005-2015 as fraction per cell. (a) ESA-CCI evaluation data (b) Simulated FPC by LPJmL4-SPITFIRE using the NI$_{orig}$ version (c) Simulated FPC by LPJmL4-SPITFIRE using the VPD$_{optim}$ version (d) Simulated FPC by LPJmL4-SPITFIRE using the NI$_{optim}$ version

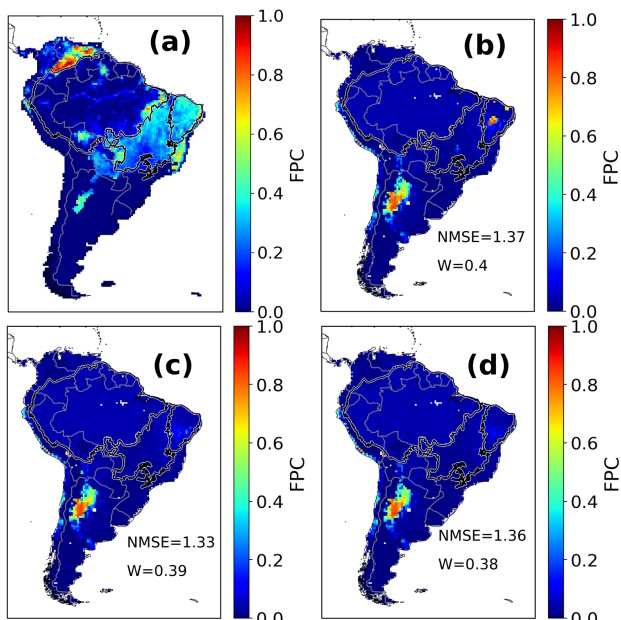

**Figure A2.** Annual FPC cover by tropical herbaceous PFT over a mean from 2005-2015 as fraction per cell. (a) ESA-CCI evaluation data (b) Simulated FPC by LPJmL4-SPITFIRE using the $NI_{orig}$ version (c) Simulated FPC by LPJmL4-SPITFIRE using the $VPD_{optim}$ version (d) Simulated FPC by LPJmL4-SPITFIRE using the $NI_{optim}$ version

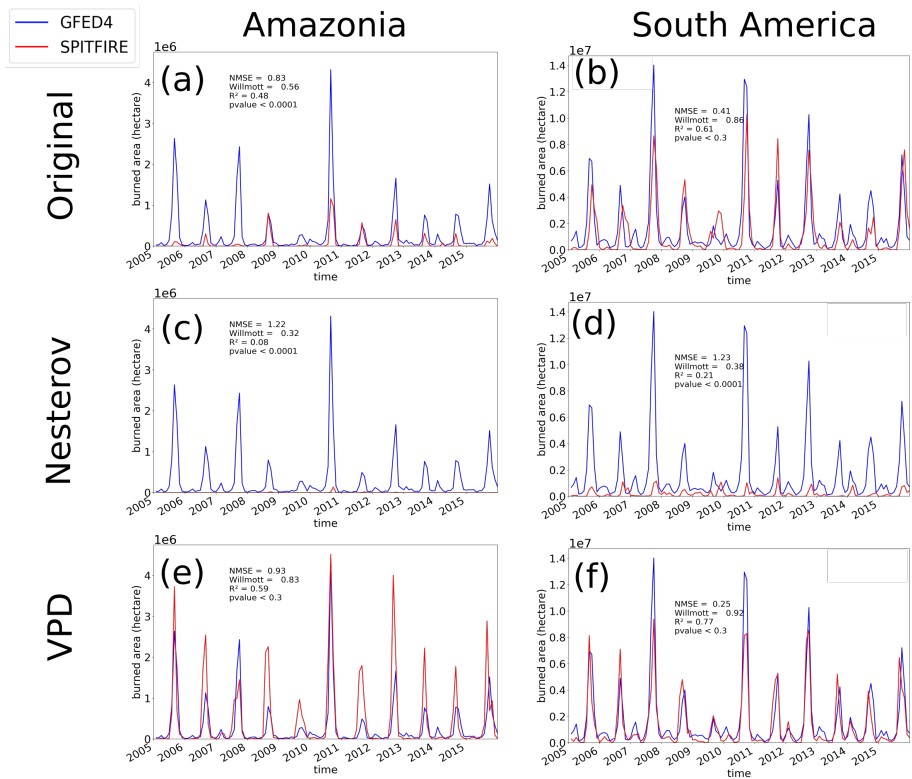

**Figure A3.** Time-series of monthly burnt area from 2005 - 2015 simulated by SPITFIRE (red lines) compared to GFED4 evaluation data (blue lines) for: (a) The Amazonia region, using $NI_{orig}$. (b) Total South America, using the $NI_{orig}$. (c) The Amazonia region, using $NI_{optim}$. (d) Total South America, using $NI_{optim}$. (e) The Amazonia region, using $VPD_{optim}$. (f) Total South America, using $VPD_{optim}$.

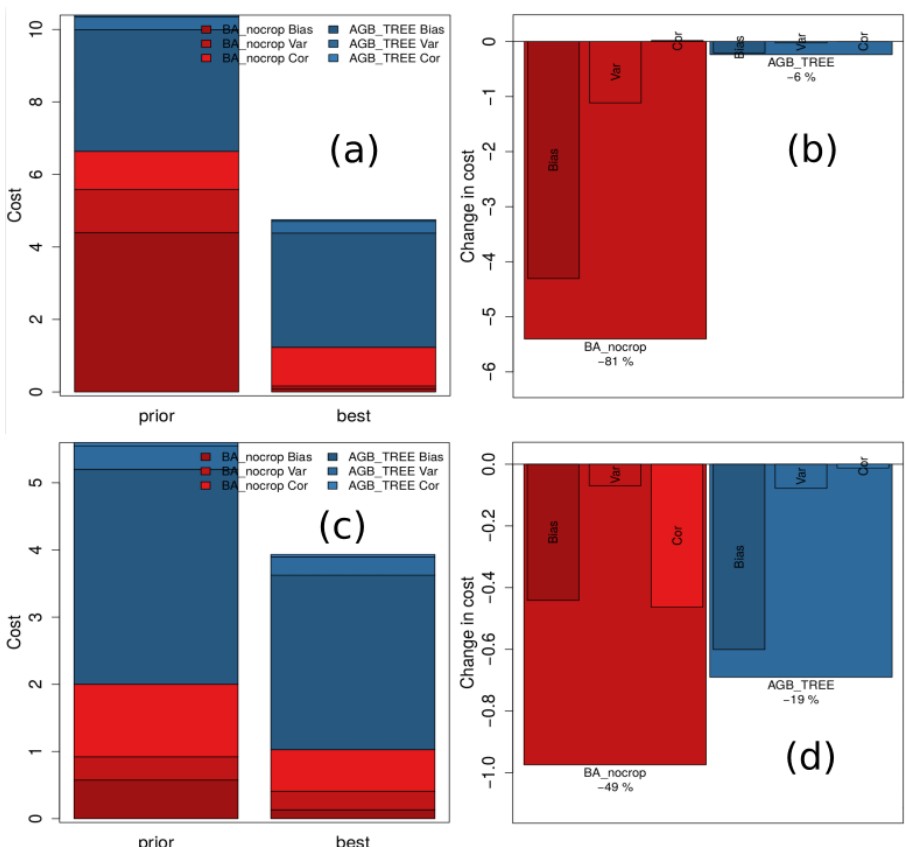

**Figure A4.** Cost reduction of the burnt area and the biomass during the optimization process, by showing the various components of the cost that are related to model-data bias, variance ratio and correlation. The cost for burnt area for $NI_{optim}$ decreased by ca. 81%, whereas the cost of the biomass only decreases by ca. 6% (a and b). For $VPD_{optim}$ the cost decreased by ca. 48% for burnt area and ca. 19% for the biomass (c and d). Hence the impact of the optimization process on burnt area is much larger due to the focus on fire parameters.

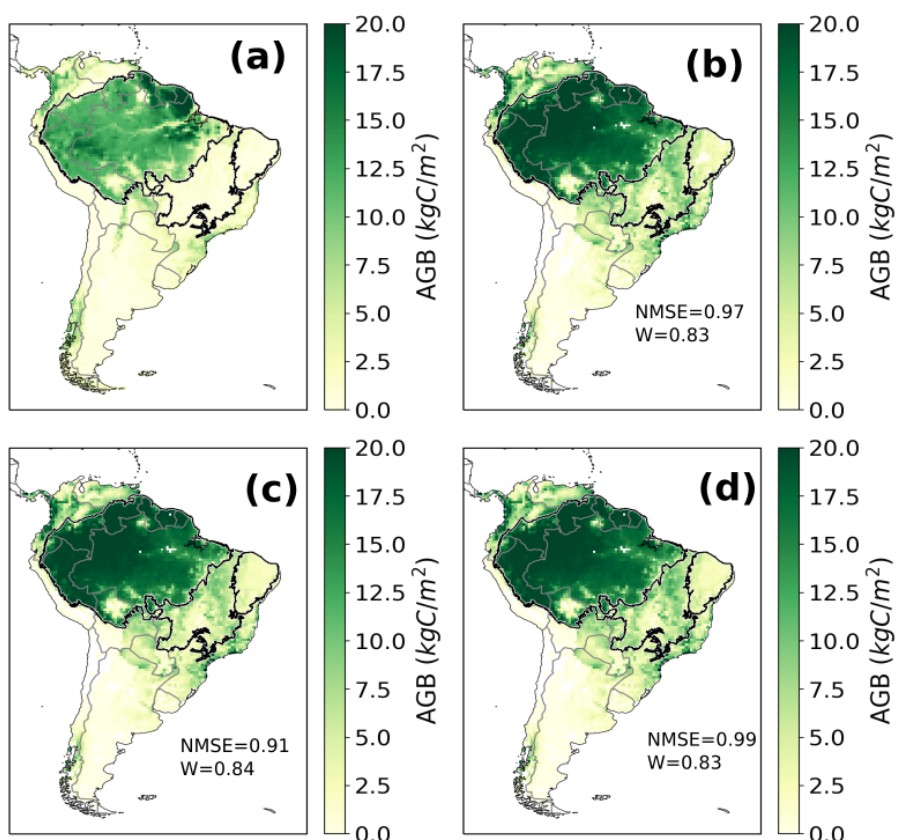

**Figure A5.** Annual above ground biomass (AGB) of trees over a mean from 2005-2015 in kgC/m$^2$. (a) Avitabile evaluation data. (b) Simulated AGB by LPJmL4-SPITFIRE in the NI$_{orig}$ version. (c) Simulated AGB by LPJmL4-SPITFIRE in the VPD$_{optim}$ version. (d) Simulated AGB by LPJmL4-SPITFIRE in the NI$_{optim}$ version.

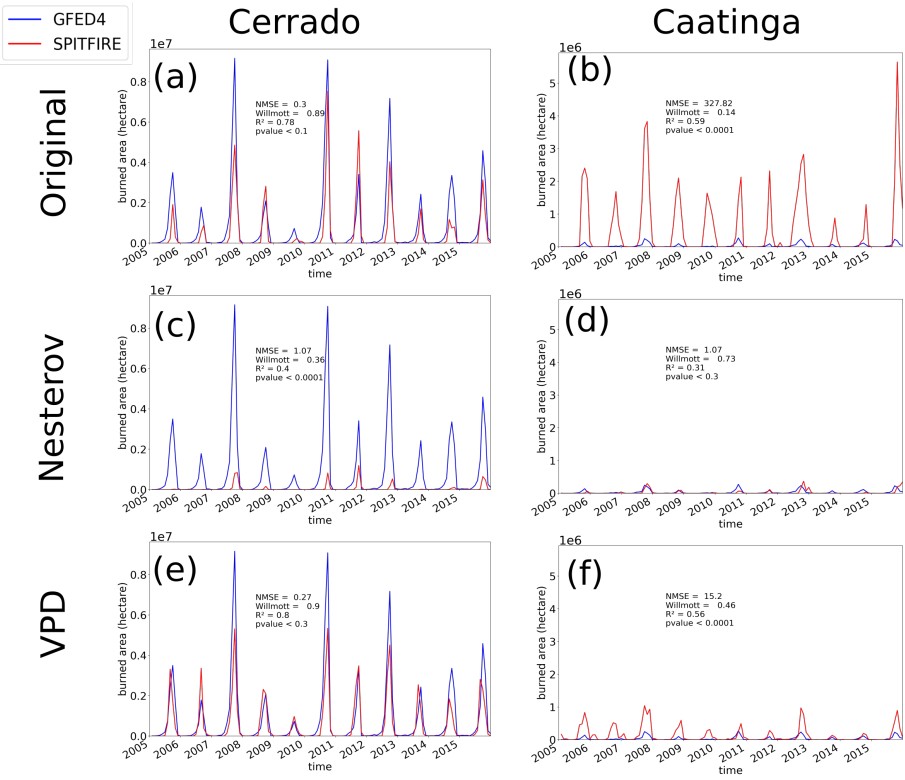

**Figure A6.** Time-series of monthly burnt area from 2005 - 2015 simulated by SPITFIRE (red lines) compared to GFED4 evaluation data (blue lines) for: (a) The Cerrado region, using $NI_{orig}$. (b) The Caatinga region, using the $NI_{orig}$. (c) The Cerrado region, using $NI_{optim}$. (d) The Caatinga region, using $NI_{optim}$. (e) The Cerrado region, using $VPD_{optim}$. (f) The Caatinga region, using $VPD_{optim}$.

*Author contributions.* MD, MF and KT designed the study in discussion with MC, MB, BS and JK. MD and MF implemented the model-data integration framework for LPJmL4. MD and WvB implemented the new fire danger index. MD performed the analysis with inputs from MF. MD wrote the paper with inputs from all Co-authors.

*Competing interests.* The authors declare that they have no conflict of interest.

5   *Acknowledgements.* This paper was developed within the scope of the IRTG 1740 / TRP 2015/50122-0, funded by the DFG / FAPESP (MD und KT). MC acknoledges the support from the projects FAPESP 2015/50122-0 (São Paulo Research Foundation), and CNPq 314016/2009-0 (Brazilian National Council for Scientific and Technological Development). MF acknowledges funding through the TU Wien Wissenschaft-spreis 2015. MB acknowledges the support of the Brazilian Research Network on Global Climate Change (Rede Clima) and of the National Institute of Science and Technology for Climate Change Phase 2 under CNPq, Grant 465501/2014-1, FAPESP, Grant 2014/50848-9 and the

National Coordination for High Level Education and Training (CAPES) Grant, 16/2014. KT and BS acknowledge funding from the BMBF- and Belmont Forum-funded project "CLIMAX: Climate Services Through Knowledge Co-Production: A Euro-South American Initiative For Strengthening Societal Adaptation Response to Extreme Events", Grant no. 01LP1610A.

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
