# Peer review of "Improving the LPJmL4-SPITFIRE vegetation-fire model for South America using satellite data"

_Geoscientific Model Development, 2019_

## Referee Comment (RC1) · Anonymous Referee #1 · 27 May 2019

(See attached PDF)

Please also note the supplement to this comment:
https://www.geosci-model-dev-discuss.net/gmd-2019-92/gmd-2019-92-RC1-supplement.pdf

---

## Referee Comment (RC2) · Anonymous Referee #2 · 14 Jun 2019

**Review of**

**"Improving the LPJmL4-SPITFIRE vegetation-fire model for South America using satellite data"**

**Drüke et al. GMD-2019-92**

The paper utilises a genetic optimization algorithm and a revised fire danger index to improve the representation of burnt area and biomass in the LPJmL4-SPITFIRE model compared to satellite-derived datasets, optimised against those same datasets. The authors also benchmarked the fractional cover of one PFT and claimed improvements to PFT distribution and temporal dynamics both (inter-annual variability and seasonal patterns). They also advocate the use of such methods for improving fire-vegetation models in general.

Investigating alternatives to the Nesterov Index in SPITFIRE (and other global fire models) and using optimisation algorithms to develop DGVMs and fire models are laudable aims and this work makes useful contributions in these directions. Simultaneously using both burnt area and biomass observations to constrain the model parameters, and the application of rigorous benchmarking metrics are also to be commended. Many parameters in SPITFIRE are very poorly constrained, so this is a promising approach to improve the model.

However, I do have substantial reservations regarding the presentation and, to some extent, the methodology, which I believe need to be addressed prior to publication. I first list my main concerns, and then a series of comments to the text. I feel confident that these concerns can be addressed in a revised version of the manuscript, perhaps with some additional analysis.

**Main concerns**

1. Whilst the optimisation procedure produces very reasonable results in the case of the VPD FDI, the Nesterov Index results are not so clear cut and cast some doubt on the efficacy of the method. Yes, the summary metrics for spatial BA do get better (at least the NSME does, the Willmott coefficient goes down, which I assume means worsening agreement?), the temporal metrics do improve drastically for Caatinga but worsen for the Cerrado, and the biomass results are basically unchanged. So that is a mixed bag. But, most critically, a visual inspection of the BA produced shows a massive reduction in fire and almost complete spatial mis-match compared to the observations, not the preferred behaviour of a fire model! There is much to discuss here which is missing from the manuscript. Benchmarking/optimising burnt area is hard due to the large amount of zero values and then high peaks, and so getting a fire peak wrong by one or two gridcells is heavily penalised. Thus an optimisation will tend towards a conservative 'no fire strategy'. This appears to be what is happening here, but is not discussed. This obviously raises questions about whether or not BA can effectively be used in such a context when it produces results which objectively (in terms of metrics) are perhaps better, but somewhat subjectively may not actually produce a more useful model.

2. The optimisation to both BA and biomass is definitely a good idea, and as far as I can tell combining the two KGE metrics is reasonable. However, as part of the paper is to

demonstrate this approach, I think there must be more discussion and analysis of this method. In particular, can the authors disentangle the relative constraints of each dataset in the method? I think this is important information for such a method. If all else fails, perhaps simply running the optimisation for BA and biomass individually would be an option.

3. No specific information on how the gridcells used in the optimisation were selected. It seems to have been done just by 'picking some'. By the authors' own admission this may bias the optimisation. Could they justify their choice a little better? Furthermore, could it be possible to run with random gridcells every time? Or gridcells close to the meteorological stations used in the preparation of the climate data? A more concrete method for select the gridcells, or at least a clearer justification, is required.

4. There is no discussion of what the optimised parameters mean in terms of process understanding or what the newly introduced 'alpha' for the VPD FDI really means. Many of the existing parameters move very little (perhaps a little surprising but also perhaps reassuringly), but the rCKs are very interesting. For $NI_{optim}$ these converge to very similar values and move away strongly from their initial values. Having similar crown kill probability for raingreen and evergreen trees flies in the face of the assumptions in SPITFIRE so far. But for $VPD_{optim}$ the story is somewhat different, with rCH for TrBE remaining very high, but rCK for TrBR also increasing. Please discuss these results, including some ecological context. And regarding the new 'alpha', what does this really mean? The very different value for TrBE compared to TrBR and TrH definitely deserves some discussion as it appears to be integrating some new factor into the FDI which the NI does not include and is not adequately represented in the other SPITFIRE PFT-specific parameters. Some discussion, even if it is a little speculative, is necessary here. In generally I can see no problem in tuning process-based models with 'black box' optimisation procedures and somewhat unphysical variables, but there must be at least some attempt to interpret and relate the results back to the processes.

5. Again, relating to the process-understanding, plots of the fire intensity resulting from the methods should be shown (possibly in an appendix if necessary). The "fuel moisture -> combustion completeness -> fire intensity -> mortality" link is a critical pathway in these results, it should be discussed explicitly but is not.

6. There is no benchmarking of the PFTs that we expect to be effected by fire! The inclusion of TrBE PFT FPC is great, but what about TrBR and TrH? These should be at least plotted, and ideally benchmarked. If the ESA CCI dataset does not have useful classifications in this regard, at least MODIS VCF MOD44B Tree-Nontree-Bare would provide some reference data for the Caatinga and Cerrado.

**Specific comments to the text**

**Abstract**

'partly poor'
 -rephrase

'as a starting point'
 – rephrase, this is not the first work to improve fire in DGVMs

'improves simulation of … plan functional type'
– is that really demonstrated?

**Introduction**

P2 *'recent decline in global burnt are*a'
– now contested (indeed by one of the authors)

P2 *'Especially in South America, tropical forests, woodlands and other ecosystems are vulnerable to increasing fire danger and land use change'*
– reference?

**Material and Methods**

P6 *'SPITFIRE further includes a surface intensity threshold'*
– please state this threshold here.  I realise that this is in the Table 2 but the units are not given.

P7 *'The fire danger index is scaled by a PFT-dependent constant, $\alpha_i$, over the number of PFTs n (Thonicke et al., 2010)'*
- in the original Thonicke et al.  2010 implementation, the $\alpha$ varied over fuel classes (ie. 1hr, 10hr, 100hr, 1000hr and live grass fuels) *not* PFTs.  Please explain and justify this change.  Also, there no mention of live grass fuels.  Are they parameterised as in original SPITFIRE?

P8 *'and a monthly mean for R to avoid unrealistic high flammability fluctuations in time steps with isolated events of very low R'*
– can the authors justify this further?  I know it is stated in the Pechnoy and Shindell paper, but it is not immediately clear why flammability fluctuations due to rainfall events should be 'unrealistic'. Perhaps with their experience with this method, the authors can provide a more convincing argument.

P8  *'Hence, we scaled the VPD up with a PFT-dependent scaling factor $\alpha_i$'*
– since this has a very different physical meaning than the $\alpha_i$ above, I strongly suggest using a different symbol.

P8 *'The general behaviour of the two indices as modelled by LPJmL in dependence of relative humidity and temperature is shown in Fig. 3'*
– Fig 3 is a nice plot, but please explain in a little more detail how the panels are comparable , as in how was the effect of vegetation taken in to account in the lower panel for VPD FDI?

P8 *'We regridded and aggregated the data set to the LPJmL resolution of 0.5 ◦ × 0.5 ◦ and to a daily time step'*
– normally climate data is the limiting factor when it comes to spatial resolution in DGVMs.  Is there any reason that the authors chose to aggregate this rather that use 0.25 degree?  Especially when the evaluation data sets are available at 0.25 degree or finer.  It seems like throwing away information.

P10 *'The optimization was performed for 40 grid-cells in South America to represent a variety of fire regimes (Fig. 2). Most of them were selected in active fire regions, especially in the Cerrado and Caatinga. In addition a few pixels with no or almost no fire occurrence (e.g. central Brazilian Amazon) were chosen.'*
– this is a rather vague description of what may be a very important choice in the optimisation procedure! See my main concern above. Please give more details in the logic here.

P10 –Despite being important the $FDI_{NI}$ and the Rothermal equations, and being poorly constrained, moisture of extinction was not mentioned as a possible parameter for optimisation. Could the authors discuss this?

P11 *'NMSE'*
– can the authors justify their choice of NMSE over NME?

P11 *'Willmott coefficient'*
– please explain its range and meaning, as is done for NMSE.

**Results**

P12 *'mainly by shifting much of the simulated burnt area from the sparsely vegetated Caatinga towards the Cerrado region'*
– this is true to some extent, but it also much is moved into Amazonia in regions where very little fire is observed in reality. In order to back up this statement, a table with the burnt area in each region for each simulation should be provided. The over-estimation of fire in Amazonia should be discussed in the Discussion section.

P13 *Figure 5*
- There is some fire in the Amazonia region, both in the data *and* in the simulations. Therefore, this region should be included in Figure 5 and Table 1, and discussed.

P13 *'Here, the TrBE showed the largest value (22.41), ca. 20 times as large as the TrBR (1.21) and TrH (1.13) (Tab. 2)'*
– there is no discussion of what this actually means in the Discussions section, please include an interpretation.

P17 *'... but also here we got an even larger improvement, when only the fire-prone regions Cerrado or Caatinga are considered (Tab. 3)'*
- Caatinga results are not shown in Tab. 3, although I think they should be. Possibly also results for Amazonia (see above)

P 18 *Figure 7*
– difference plots are great and I can see the logic behind including the difference relative to the original model version to show improvements (as you have done) but please show the absolute values too (as in Figure 4).

Discussion

P19 *'Another result of the optimizing procedure, using FDI_VPD , was the improvement of the PFT distribution..'*
– I am not sure that statement is justified given the very small improvement in TrBE and no demonstrated improvement in the other PFTS.

P19 '*it emphasizes that three parameter sets determining PFT distribution*'
 – what three parameter sets?  You mean three PFTs? Or something else? Please clarify.

P20  '*Limitations during the optimization process*'
– this heading is somewhat confusing and maybe should better be 'Limitations of the optimization process'

P20 '*As shown in Fig. 8, the modelled PFT coverage showed an equal distribution of tropical raingreen and evergreen PFTs throughout wide parts of central-northern South America*'
– Fig. 8 shows no such thing, it only shows the FPC of the evergreen PFT.  Of course, it may simply be that the caption is incorrect somehow, but otherwise the distribution of the raingreen PFT must be shown to demonstrate this.

P20 '*By choosing a large amount of optimization cells in the, by NI orig , strongly overestimated Caatinga region, the burned area decreased there significantly after the optimization*'
 – this (slightly confusing statement) would appear to indicate that the authors acknowledge that their results depend heavily on the choice of gridcells for the optimisation (see above)

P20  '*In the Cerrado and especially the Caatinga, however, trees suffer from water stress in the dry season and should shed their leaves to avoid mortality related to drought or growth efficiency. The resulting dominance of the TrBR PFT has a very different effect on fire spread and is more fire-tolerant (different fuel characteristics and resulting fire intensity), thus has a lower fire-related mortality.*'
– whilst this a reasonable enough statement (in fact pretty much inherent in the construction of DGVMs and SPITFIRE) it is hard to see what it has to do with the limitations of the optimisations process.

P21 – '*Nonetheless, we were able to improve the interannual variability and hence, the model performance during extreme years for the Cerrado and Caatinga regions (e.g. for 2007/2008, Fig. 5).The optimized SPITFIRE is now able to model accurately the climate dependent seasonal and interannual variability as well as the spatial extent of fire on natural land throughout the fire-prone woodlands of South America.*'
 – yes and no.  In the Cerrado the results from Fig 5. are not significantly different between VPD and Original, and whilst the results are better in the Caatinga for VPD, most of this comes down to the overall normalisation, it is hard to see if VPD really catches between IAV and seasonal dynamics.  In fact, the $R^2$ (which is insensitive to the normalisation) actually gets worse going from Original to VPD.  So these statements need much more nuance.  And a plot of the normalised time series (equivalent to Fig 5., at least for the Caatinga) might be a more effective way showing improvements in IAV and seasonal dynamics.

P21  *entire section titled 'Outlook*
 - the way ahead in improving fire modules in DGVMs' – this text does not really fit the title.  Much of it refers specifically SPITFIRE or LPJml, specifically their current limitations.  Please reconsider/revise/re-title this section.

P21  The statements '*it would be possible to use an even more comprehensive fire danger index (e.g. Canadian Fire Weather Index; Wagner et al., 1987) or different fire danger indices for different biomes*' and '*In a global modelling approach, however, we need to find one fire danger index*' seem to contradict each other, please resolve!

**Conclusions**

P21 *'We have demonstrated a major improvement of the fire representation within LPJmL4-SPITFIRE by implementing a new fire danger index and applying a model-data integration setup to optimize fire-related parameters.'*
- whilst there are tangible improvements, they are only tested and in the Caatinga and Cerrado, the region for which the optimisation was done (which you do mention in the next sentence). I would suggest toning this down slightly.

P21 *'We improved the seasonal and interannual variability'*
 – I have yet to be convinced of this, especially as the $R^2$ for the time series are not improved with VPD. And I am not sure how to interpret the Willmott coefficient as this is not described.

P21 *'A realistic representation of fire is also crucial for fire-vegetation-climate feedbacks and is hence necessary for DGVMs coupled within and comprehensive Earth system model.'*
 – I think you can drop that sentence, as it attempts to summarise and justify fire modelling in general rather than this work. The penultimate sentence is fine to end with.

---

## Author Comment (AC1) · 9 Aug 2019

**Reviewer 1**

*We thank reviewer 1 for the detailed and thorough comments. Our replies to the comments are inserted below in blue colour.*

Reviewer 1

Review: Improving the LPJmL4-SPITFIRE vegetation-fire model for South America using satellite data

General comments
Process-based global fire models are widely considered critical components of dynamic global vegetation models, with certain biomes—especially tropical and subtropical savannas and grasslands—being strongly regulated by fire disturbance. However, many such fire models have been developed based on parameterizations from extratropical biomes. In this manuscript, Drüke et al. use an automated technique to reparameterize a global fire model to improve its performance in the Brazilian Caatinga and Cerrado biomes with regard to both burned area and biomass. The authors perform this optimization using actual runs of the vegetation model rather than in some kind of offline mode—something that has only rarely been done before for fire models, but which could be a valuable component of the global vegetation-fire modeling toolbox. This, combined with the fact that the authors describe their methods thoroughly and walk through the results in a logical manner, lead me to recommend that this manuscript be accepted for publication pending minor revisions.

We thank the Reviewer for this positive feedback.

Specific comments

•P2 L13: The authors describe the Caatinga as fire-prone, but the referenced map (Fig. 1) does not provide much support for that assertion. The authors should clarify in the text what they mean.

We agree that the term fire-prone is not strictly applicable for the Caatinga region. Without human influence on vegetation composition and fire, the Caatinga biome is not prone to fire. Fire risk has increased with human influence. We wanted to stress the fact, that fire occurrence of the Caatinga in the prior SPITFIRE implementation was too large,  because dead and live fuel was overestimated in the model. It is one of the achievements of this study to reduce this model error. We now make a better distinction between fire-ecological conditions of the biome and what is captured in the model. We therefore changed the wording accordingly to only describe the Cerrado as fire prone in P2 L12-13:

*"This study focuses on the fire behavior in central-northern South America and especially on the Brazilian biomes Caatinga and Cerrado, which is the most fire-prone region in South America (Fig. 1)"*

•P6 L1: "their effectiveness to ignite a fire is 0.04" is unclear. Better would be something like, "4% of cloud-to-ground strikes can start a fire."

We followed the Reviewer's suggestion and express these numbers now in percent in P6 L1:

*"assuming that 20 % of the flashes reach the ground and 4 % of cloud-to-ground strikes can start a fire."*

•P6 L9:
As far as I can tell, this is the first time the parameter named p_d " has been used in regard to SPITFIRE. I suggest using some other symbol, as this p_d " could be easily confused with P_d (population density) from Thonicke et al. (2010).

Thanks for pointing to this inconsistency in naming our model variables. We have corrected it and changed it to p_t, where the t stands now for time and not for the duration (d).

"per day" is misleading; SPITFIRE as described in Thonicke et al. (2010) does not allow for multi-day fires, and thus this is simply the maximum fire duration.

Indeed, SPITFIRE does not directly allow multi-day fires. The maximum fire duration is 240 minutes. Since the model time-step is however one day this time duration limits the fire to 240 minutes for each time-step. SPITFIRE does not model individual fires, but assumes the burning conditions for all fires ignited on the same day to be similar. If burning conditions are comparable the next day, new ignitions would be computed resulting in comparable daily area burnt. Hence, to our opinion the wording in this case is correct.

•P11 L1–2, P21 L21–27: Because LPJmL does not allow fire on managed lands, the authors exclude cropland burning from the observed data in their comparisons. This is reasonable, but ignores the fact that a fair amount of Cerrado is actually used as pasture, primarily in the southern part of the region (Sano et al., 2010; Parente et al.,2017). I don't think this makes a huge difference in the context of this manuscript, because (a) only a few of the 40 sampled grid cells were from the southern Cerrado, and (b) the main takeaway from this paper should be the use of the optimization algorithm, rather than the exact parameter values it gives. However, the authors should (briefly) address this issue in the text.

We acknowledge the fact that excluding cropland in the optimization process is not optimal. Unfortunately we did not have a dataset, which also excludes pastures. This leads to a slightly wrong burnt area in the optimization process. We already addressed this issue in P24 L26-29:

*"We do, however, not account for the proportion of grassland, which is used as pastures for e.g. cattle ranching. Since in SPITFIRE fire is not enabled on managed grassland, our results show a slightly smaller fire amplitude throughout South America than could be expected with pastures included and hence also compared to the GFED4 evaluation data set."*

We changed in the above paragraph the term "grassland" to "pasture" and included a sentence about this issue in P11 L32–34:

*"As LPJmL does not simulate fire on managed lands, we excluded burnt area on cropland classes from model-data comparisons. Due to lack of data we however did not account for the proportion of pastures."*

•Sect. 2.4 and/ or Sect. 3.2: For the benefit of other researchers interested in using this or a similar optimization algorithm, it would be helpful to know various pieces of info about the process. How many model runs were required? How long did they each take? How was the decision made to halt the optimization—was it manual, or did the algorithm reach a stop condition? If the latter, what was/were the stop condition(s)? Etc. This level of technical detail is more than appropriate for GMD.

We agree that some more technical details are important to the GMD reader and we therefore

added the following paragraph in Sect. 2.4 (P11 L5-17):

*"In genetic optimization algorithms, each model parameter is called an individual with a corresponding fitness, which represents the cost of the model against the observations. At the beginning of the optimization process, the GENOUD algorithm creates a generation of individuals based on random sampling of parameter sets within the prescribed parameter ranges. After the calculation of the cost of all individuals of the first generation, a next generation is generated by cloning the best individuals, by mutating the genes or by crossing different individuals (Mebane and Sekhon, 2011). This results after some generations in a set of individuals with highest fitness, i.e. parameter sets with minimized cost. To find an optimum parameter set we were also using the BFGS gradient search algorithm (named after the authors Broyden, 1970; Fletcher, 1970; Goldfarb, 1970; Shanno, 1970) within the GENOUD algorithm. An optimized parameter set of the BFGS algorithm is used as individual in the next generation. We were applying the GENOUD algorithm with 20 generations and a population size of 800 individuals per generation, which corresponds to 16000 single model runs. We decided on this amount of iterations, because the cost kept almost constant in the last iterations and the parameter values did not change to the 6th digit, beyond which changes are not really relevant for model applications. During the optimization we ran the model parallel in for each gridcell (40 grid cells and CPU's, 3.2GHz) and had a total optimization time of ca. 24 hours."*

•Sect. 2.6: The authors do a good job describing how to interpret values of the NMSE, but they should do the same for the Willmott coefficient of agreement. What are the possible values? What are "milestone" values (e.g., for NMSE, 0 vs. 1 vs. >1)?

Thanks for pointing to this missing information. The information on the meaning of the the different values for the Willmott coefficient was indeed missing. We have now added the following sentence in P12 L17-20:

*"The Willmott coefficient is a squared index, where a value of 1 stands for perfect agreement between simulated and modelled runs and gets smaller for worse agreements with a minimum of 0. Unlike the coefficient of determination, the Willmott coefficient is additionally sensitive to biases between simulations and observations."*

•P13 Fig. 5: It would be helpful to use the same Y-axis for all subplots in the right column (subplots b, d, f), as was done for the left column.

We also thought about this issue but had decided before to not use the same Y-axis for all subplots in the right column, because the values differ by an order of magnitude. Hence, the lines in Fig. 5 f would be hardly visible. We agree with the reviewer that different scales in this case are very unusual and we apply now a logarithmic Y-axis for the Caatinga. This enables us to use the same Y-axis for all three model versions. A non-logarithmic version remains in the Appendix. We now updated Fig. 5 and added the following statement to the description of the Figure:

*"Note the logarithmic scale for the Caatinga, which was applied in order to account for the large differences between the different model versions (for a non-logarithmic version see Fig. A6)."*

[Figure]

Fig. 5. Time-series of monthly burnt area from 2005 - 2015 simulated by SPITFIRE (red lines) compared to GFED4 evaluation data (blue lines) for: (a) The Cerrado region, using NI_orig. (b) The Caatinga region, using the NI_orig. (c) The Cerrado region, using NI_optim. (d) The Caatinga region, using NI_optim. (e) The Cerrado region, using VPD_optim. (f) The Caatinga region, using VPD_optim. Note the logarithmic scale for the Caatinga, which was applied in order to account for the large differences between the different model versions (for a non logarithmic version see Fig. A6).

•P13 Fig. 5 and P16 Fig. 6: Nesterov and VPD rows should be swapped, since in the rest of the paper the Nesterov Index is usually discussed first.

We swapped the Nesterov and the VPD rows, as suggested by the Reviewer.

•P16 Fig. 6: The use of lines here is confusing, since that usually implies some kind of change over time. The authors should seriously consider using a bar graph here instead.

We thank the reviewer for the suggestion to change Fig. 6.

We completely remade Fig. 6 by excluding the lines and changing the X- and Y-axis. We also now show each parameter individually. To our opinion the use of a bar plot would limit the information about the exact uncertainty for each parameter for the different PFTs. We hope that we have addressed the reviewer's concern adequately with this new figure.

[Figure]

Fig. 6. Relative uncertainty of model parameters after optimization for (a) NI_optim and (b) VPD_optim. The relative uncertainty is the ratio of the uncertainty after the optimization (range of all parameter sets with low cost, below the 0.05 quantile) divided by the uncertainty before the optimization (range of the parameters for the optimization). Low and high values of relative uncertainty indicate strongly and weakly constrained parameters, respectively. SIT denotes the surface intensity threshold

•P17 L3–4: "The model optimization scheme focuses on fire parameter [sic], hence the model performance can only improve in fire-prone biomes, i.e. not in, e.g., wet tropical forest where fire is absent." This is not strictly true. Model performance could improve in wet tropical forest if the initial parameterization (a) performed badly there

with regard to burned area (i.e., simulated almost any fire at all) and (b) underestimated biomass. It just so happens that neither of these conditions are met by the initial LPJmL configuration. This may seem like a minor quibble, but it could mislead other researchers interested in applying this or a similar optimization algorithm to their own models. It is important to be clear that optimizing a fire model can improve performance with regard to vegetation parameters not necessarily where fire is frequent, but rather where fire is modelled poorly.

We agree that this statement was a bit misleading. We wanted to emphasize that we did not get an improvement in AGB and FPC in areas not affected by fire, both in the model and in the evaluation data, and hence it does not contribute to the improvement due to the optimization process. We now clarify this issue by changing the quoted sentence into (P19 L5-10):

*"The model optimization scheme focuses on fire parameters, hence the model performance for AGB can only improve in areas, where the fire occurrence has been modelled poorly and the vegetation-fire interactions have improved due to the optimization process. For example in the center of the Amazon rainforest almost no fire is found in the evaluation data nor is simulated. Hence no improvement of burned area as well as AGB can be achieved. On the other hand, in regions where the modelling error of burnt area is now reduced, this can also improve simulated AGB, hence vegetation-fire interactions."*

•P20 L7–8: Presumably the authors are making this assertion based on the fact that the indicated region is modelled as ~50% tropical evergreen, but Fig. 8 does not appear to say anything about the tropical raingreen PFT—just evergreen. The authors should clarify this.

We agree with the reviewer that readers not familiar with DGVMs or specifically LPJmL should be provided with such information for clarity. We new added the maps of the FPC of tropical raingreen and herbacious C4 PFTs to the supplement (Figure A1 and A2) and added a short description to the result section now for all 3 tropical PFTs in P20 L5-8:

*'Also for the TrBR and TrH PFT distributions the optimization lead to an improved performance using the VPD_optim in the Caatinga and Cerrado, whereas the PFT distribution in the Amazon remained similar to the prior PFT distribution. In the NI_optim version, parameter optimization only slightly reduced TrBR cover showing a worse performance compared to VPD_optim. However, herbaceous cover changed only slightly in all optimization experiments (Fig. A1 and A2).'*

[Figure]

Fig. A1. Annual FPC cover by tropical broadleaved raingreen PFT over a mean from 2005-2015 as fraction per cell. (a) ESA-CCI evaluation data (b) Simulated FPC by LPJmL4-SPITFIRE using the NI_orig version (c) Simulated FPC by LPJmL4-SPITFIRE using the VPD_optim version (d) Simulated FPC by LPJmL4-SPITFIRE using the NI_optim version

[Figure]

Fig. A2. Annual FPC cover by tropical herbaceous PFT over a mean from 2005-2015 as fraction per cell. (a) ESA-CCI evaluation data (b) Simulated FPC by LPJmL4-SPITFIRE using the NI_ orig version (c) Simulated FPC by LPJmL4-SPITFIRE using the VPD_optim version (d) Simulated FPC by LPJmL4-SPITFIRE using the NI_optim version

•P21 L23–27:
How do the authors reach the conclusion that including fire on managed land would increase "fire amplitude" (this phrase should be reworked, by the way) and improve interannual variability? Why might it not also (or instead) improve annual mean?
Citations should be added regarding the real-life use of fire on managed lands (e.g., Laris, 2002).
Citations should be added regarding the simulation of fire on managed lands (e.g., Pfeiffer et al., 2013; Rabin et al., 2018).

We agree that this statement is relatively vague. We meant to say that we would have a slightly larger burnt area in regions, which also include pastures. The larger burnt area would cause a

larger annual total (or mean) burnt area and also likely a larger seasonal amplitude (if there are no fires during the wet season). Also the overall model performance would improve and not just the interannual variability. However, we expect this effect to be relatively small, since only a few grid cells are covered by a substantial fraction of pasture land in the study area. . We thank the reviewer for the valuable suggestions, including citations, and rewrote this section as following (P24 L27-33):

*"Since in SPITFIRE fire is not enabled on pasture, our results show a slightly smaller burnt area throughout South America than could be expected with managed land included and hence also compared to the GFED4 evaluation data set. This effect is however small, because pasture lands cover a substantial fraction only in very few grid cells (e.g. southern Cerrado; Parente et al., 2017). Fire on managed land is generally difficult to predict in a DGVM because the reason and timing of using fire depends less on climatic factors but mostly on social and political decisions which can vary between countries, regions and localities. We expect further improvement of model performance especially in regions of large land-use areas with fires on pastures included (e.g. Rabin et al., 2018; Pfeiffer et al., 2013)."*

•P22 L1–2: This is incorrect; Rabin et al. (2018) did indeed optimize FINAL.1 within a dynamic global vegetation model (LM3). Also, the authors should (at least briefly) discuss the pros and cons of their method relative to the one used by Rabin et al. (2018); this would be valuable for other researchers interested in optimization methods.

We thank the reviewer for noting this error. Rabin et al. (2018) did indeed optimize FINAL.1 within a DGVM. We now refer to this study and explain the major differences to the approach used in our study. As suggested we added a paragraph about the pros and cons of our method relative to the one us used by Rabin et al. (2018) in the discussion (P23 L27 - P24 L8):

*"Systematic optimizations within a model-data integration setup of fire models which are embedded in a DGVM are still very rare. Previously, Rabin et al. (2018) optimized the fire model FINAL.1 within the land-surface model LM3. Our study differs from Rabin et al. (2018) in the conceptual design of the vegetation-fire models and the optimization process. While LM3 has been run on a 2° longitude by 2.5° latitude, running LPJmL at 0.5° by 0.5° grid cell resolution allows us to account for spatial differences in climate , vegetation and fire interaction. While FINAL.1 is a process-based model, many calculations (e.g. the fire spread routine) are done by multiplying the important factors and fitting the resulting values to observational data. SPITFIRE tries to model the important fire variables by simulating the underlying processes, and by taking the influence of climate and the different fire ignitions into account. An advantage of FINAL.1 is the inclusion of agricultural fires based on a statistical approach. Whereas Rabin et al. (2018) used a local search algorithm (Levenberg-Marquardt algorithm) to optimize their model, we used a global search algorithm (genetic optimization). Local search algorithms depend on the chosen initial parameter sets and might eventually end up in a local optimum. A genetic optimization algorithm allows to explore the full parameter space and hence gives a higher chance to find the global optimum. However, local search algorithms require less iterations than global search algorithms (300 in Rabin et al. (2018) vs. 16000 in our study). Forkel et al. (2014) tested the optimization of LPJmL with different optimization algorithms and found that it was not feasible to optimize LPJmL with a local search algorithm. Rabin et al. (2018) ran the model during the optimization process only for the period of 1991-2009, whereas in our optimization setup we made complete model runs including 5000 years of spinup in LPJmL in order to get a model equilibrium for each tested parameter combination."*

•P22 L14–15: The authors address availability of the model code, which presumably refers to LPJmL. But what about the genetic optimization code?

We will publish the code of LPJmLmdi along with the model code on the github page of LPJmL: https://github.com/PIK-LPJmL/LPJmLmdi.

Technical corrections
• P9 L5: "form" should be "from"
• P10 L22: "dependend" should be "dependent"
• P11 L19: "simulations" should be "simulation"
• P11 L25: "Caating" should be "Caatinga"
• P17 Table 3: "Evergreem" should be "Evergreen"
• P17 L3: "parameter" should be "parameters"
• P18 L12: "significante" should be "significance"
• P18 L13: "particular" should be "particularly"
• P19 L6: "particular" should be "particularly"
• P21 L12: "seperate" should be "separate"

We thank the reviewer for the detailed and focused review of our manuscript. We have applied all technical corrections as suggested by the reviewer.

[revised manuscript text omitted]

---

## Author Comment (AC2) · 9 Aug 2019

**Reviewer 2**

*We thank Reviewer 2 for the detailed and thorough comments. Our replies to the comments are inserted below in blue colour.*

Review of
"Improving the LPJmL4-SPITFIRE vegetation-fire model for South America using satellite data"
Drüke et al. GMD-2019-92
The paper utilises a genetic optimization algorithm and a revised fire danger index to improve the representation of burnt area and biomass in the LPJmL4-SPITFIRE model compared to satellite- derived datasets, optimised against those same datasets. The authors also benchmarked the fractional cover of one PFT and claimed improvements to PFT distribution and temporal dynamics both (inter-annual variability and seasonal patterns). They also advocate the use of such methods for improving fire-vegetation models in general. Investigating alternatives to the Nesterov Index in SPITFIRE (and other global fire models) and using optimisation algorithms to develop DGVMs and fire models are laudable aims and this work makes useful contributions in these directions. Simultaneously using both burnt area and biomass observations to constrain the model parameters, and the application of rigorous benchmarking metrics are also to be commended. Many parameters in SPITFIRE are very poorly constrained, so this is a promising approach to improve the model.
However, I do have substantial reservations regarding the presentation and, to some extent, the methodology, which I believe need to be addressed prior to publication. I first list my main concerns, and then a series of comments to the text. I feel confident that these concerns can be addressed in a revised version of the manuscript, perhaps with some additional analysis.

We thank the Reviewer for this feedback and we hope that the revised manuscript through its improved analysis and by clarification of the methods will remove the reviewer's concern .

Main concerns
1. Whilst the optimisation procedure produces very reasonable results in the case of the VPD FDI, the Nesterov Index results are not so clear cut and cast some doubt on the efficacy of the method. Yes, the summary metrics for spatial BA do get better (at least the NSME does, the Willmott coefficient goes down, which I assume means worsening agreement?), the temporal metrics do improve drastically for Caatinga but worsen for the Cerrado, and the biomass results are basically unchanged. So that is a mixed bag. But, most critically, a visual inspection of the BA produced shows a massive reduction in fire and almost complete spatial mis-match compared to the observations, not the preferred behaviour of a fire model! There is much to discuss here which is missing from the manuscript. Benchmarking/optimising burnt area is hard due to the large amount of zero values and then high peaks, and so getting a fire peak wrong by one or two gridcells is heavily penalised. Thus an optimisation will tend towards a conservative 'no fire strategy'. This appears to be what is happening here, but is not discussed. This obviously raises

questions about whether or not BA can effectively be used in such a context when it produces results which objectively (in terms of metrics) are perhaps better, but somewhat subjectively may not actually produce a more useful model.

We thank the Reviewer for pointing out that we need to explain and discuss in more detail how the optimization was done and discuss better the implication of spatial mismatch.
Indeed, the reviewer is right that optimizing a model against burned area is challenging because #1 of the skewed statistical distribution of burned area and #2 because temporal or spatial mismatches in simulated burning can cause large model-data errors. We added a paragraph to the Discussion in P22 L19-33 to clarify and explain this issue:

*Generally, optimizing a model against burned area is challenging because 1) of the skewed statistical distribution of burned area and 2) because temporal or spatial mismatches in simulated burning can cause large model-data errors. These issues can be avoided with the choice of an appropriate cost function. For example, squared-error metrics tend to underestimate the variance of burned area in comparison to, e.g., the Kling-Gupta efficiency as it has been shown in the optimization of an empirical model for burned area (see Table A3 in Forkel et al. 2017). Here, the optimum parameter set for the Nesterov index-based model resulted in almost no fires across South America. Thereby the optimization algorithm tries to decrease the model error by tending towards a conservative 'no fire strategy' for all biomes. This result nicely demonstrates the need to evaluate model optimization results against spatially and temporally independent data and independent variables (Keenan et al. 2011).*
*The Nesterov index is not able to capture fire variability within the Caatinga as well as the Cerrado at the same time. This shows that the difference in the PFT distribution between these two biomes is not adequately modelled by LPJmL or just using PFT dependent scaling factors did not sufficiently improve the model performance when using the Nesterov index. On the other hand, using the VPD fire danger index reduced the model error for burned area in both biomes, by improving the modelled performance for the Caatinga and maintaining the good performance of the Cerrado region. Since improved performance of the fire model mainly had minor effect on improving FPC of the tropical PFTs, the presented optimization scheme has to go along with process-based improvements in both, in the fire and in the vegetation modules of LPJmL.*

Furthermore thanks to the reviewers comment we noted a small error in Figure 5: The R² of 5a (original model version for the Cerrado) has been wrongly written as 0.87. The real value, as was correctly written in the text of the first manuscript versionaper, is however 0.78. Hence we have a slight improvement by the VPD version in all three metrics. We are very sorry for the confusion caused by this spelling error.
We think that by fixing this formal error and by providing additional analysis in the revised version of the manuscript (see the answers to the other concerns raised by the reviewer below) the improvements in the performance of modelled burnt area in South America are now demonstrated better. We hope to have addressed the major concerns of the reviewer adequately.

2. The optimisation to both BA and biomass is definitely a good idea, and as far as I can tell combining the two KGE metrics is reasonable. However, as part of the paper is to demonstrate this approach, I think there must be more discussion and analysis of this method. In particular, can the authors disentangle the relative constraints of each dataset in the method? I think this is important information for such a method. If all else fails, perhaps simply running the optimisation for BA and biomass individually would be an option.

The main focus of this paper was an optimization just focussed on fire parameters. Hence, the change in biomass is relatively small and heavily depending on changes in the burnt area. We included the biomass in the optimization to make sure that the biomass would not be impacted by fire-effect processes to avoid the model performance getting worse.
To show the small improvement in the cost of the biomass, compared to the cost of the burnt area, we have now included in the Appendix a comparison of the cost reduction during the optimization process which is also shown here.

[Figure]

Fig A4: Cost reduction of the burnt area and the biomass during the optimization process, by showing the various components of the cost that are related to model-data bias, variance ratio and correlation. The cost for burnt area for NI_optim decreased by ca. 81%, whereas the cost of the biomass only decreases by ca. 6% (a and b). For VPD_optim the cost decreased by ca. 48% for burnt area and about 19% for the biomass (c and d). Hence the impact of the optimization process on burnt area is much larger due to the focus on fire parameters.

We added a comment on this and the reference to the appendix in the Results in P18 L3-6:

*The decrease in the model error (cost) due to the optimization process has been mainly due to improvement in the burnt area. While for the NI_optim the cost of the burnt area dataset improved by 81%, the cost of the biomass dataset improved just by 6%. In case of the VPD_optim the cost of the burnt area dataset improved by 49%, whereas the biomass dataset improved by 19% (Fig .A5).*

Furthermore we added to the Discussion in P21 L27-30:

*Due to the focus on fire related parameter, the cost of the burnt area dataset decreases much more than the cost of the biomass dataset (Fig. A5). Hence we only get a substantial improvement in model performance in semi-arid, fire-prone biomes, where vegetation dynamics and fire are strongly coupled.*

3. No specific information on how the gridcells used in the optimisation were selected. It seems to have been done just by 'picking some'. By the authors' own admission this may bias the optimisation. Could they justify their choice a little better? Furthermore, could it be possible to run with random gridcells every time? Or gridcells close to the meteorological stations used in the preparation of the climate data? A more concrete method for select the gridcells, or at least a clearer justification, is required.

We thank the reviewer for noting this lack of explanation of how the grid cells were selected. In this study we selected the grid cells manually. We justified our choice better by adding the following paragraph to the methods in P10 L22-31.

*The optimization was performed for 40 grid-cells in South America to represent a variety of fire regimes (Fig. 2). We selected the grid cells manually to cover active fire regions (either in the model or in the evaluation data), specifically in the Cerrado and Caatinga. We selected a high density of grid cells in the Caatinga region to improve the very poor model performance in this region.  To make sure that the model performance in the Caatinga and Cerrado was not achieved at the cost of a poor performance in other areas, we also additionally selected some cells in areas where initial fire modeling gave good results, as well as in areas where minimal or no fire occurs (central Brazilian Amazon). After inspection of the results, minor adjustments were made and the selection of the grid cells was modified to account for*

*neglected regions (which showed worsening of the model performance). These initial analyses actually demonstrate that the choice of grid cells is important for the model optimization and requires the development of a more thorough selection method in future model optimization applications.*

A random sampling of grid cells during the optimization might potentially result in more robust model parameters as similar methods are successfully used in several machine learning approaches to make more robust predictions (e.g. bagging in random forest regressions). However, such a bagging  of training points (or grid cells) within an optimization of a DGVM is currently computationally not feasible because it would require to run the optimization algorithm several (hundreds) times. Sampling random grid cells within a single optimization run will likely not result in a parameter optimum because the cost would change in each iteration which is however not related to model parameters but to the sampling of the grid cells.
We hope to have explained better why and how we manually selected our grid cells.

4. There is no discussion of what the optimised parameters mean in terms of process understanding or what the newly introduced 'alpha' for the VPD FDI really means. Many of the existing parameters move very little (perhaps a little surprising but also perhaps reassuringly), but the rCKs are very interesting. For NI optim these converge to very similar values and move away strongly from their initial values. Having similar crown kill probability for raingreen and evergreen trees flies in the face of the assumptions in SPITFIRE so far. But for VPD optim the story is somewhat different, with rCH for TrBE remaining very high, but rCK for TrBR also increasing. Please discuss these results, including some ecological context.
And regarding the new 'alpha', what does this really mean? The very different value for TrBE compared to TrBR and TrH definitely deserves some discussion as it appears to be integrating some new factor into the FDI which the NI does not include and is not adequately represented in the other SPITFIRE PFT-specific parameters. Some discussion, even if it is a little speculative, is necessary here. In generally I can see no problem in tuning process-based models with 'black box' optimisation procedures and somewhat unphysical variables, but there must be at least some attempt to interpret and relate the results back to the processes.

We thank the reviewer for this important note. While we already explain some of our parameter results in section 3.2 in terms of process understanding we now added some ecological interpretation of the VPD_optim results  in the Discussion in P22 L12-17.

*The mortality risk of TrBE for VPD_optim remains close to the prior value of 1, confirming previous assumptions about its high fire sensitivity. Whereas the rCK for TrBR increased to 0.48, close to the upper boundary of the optimization, meaning that a mortality risk of 50% when the full crown is scorched and a 7% mortality risk when 50% of the crown is scorched, which makes the TrBR less resistant against crown damage than before. Due to this changes the overestimation of biomass in the original model for the Cerrado/Caatinga region decreased (see Fig. 7).*

Moreover, we agree that the new alpha values deserve some further discussions and we now added a paragraph to the Discussion section in P22 L2-12:

*'The fire danger index scaling factors (alpha_NI_i and alpha_VPD_i) convert the quantified fire risk (NI or VPD) into the actual fire danger (FDI). Both scaling factors thus set the magnitude of the fire danger for the different PFTs. Hence they impact directly the fire spread, burnt area and the number of fires as well as indirectly fire mortality. These very important parameters vary significantly for the different PFTs. TrH has the smallest scaling factor in case of both FDIs, which leads to a lower fire danger compared to the other PFTs. This indicates a prior overestimation of the fire danger of grass in tropical South America, as grasslands are generally parametrized to have a low fire resistance and moisture content and can hence burn very easily. This overestimation, compared to tree PFTs has been decreased by the optimization. In case of the VPD also the TrBR is scaled by a much smaller factor than the TrBE, which leads to a lower fire danger index. This is due to the fact, that the TrBR is dominant in dry and fire-prone regions, which experience frequent fires. Here the burnt area was often overestimated by SPITFIRE (e.g. Caatinga or eastern Cerrado) and is now decreased. On the other hand, a larger FDI for the TrBE allows more fire in wetter regions at the edge between the Cerrado and the Amazon rainforests, where TrBE is more dominant.'*

5. Again, relating to the process-understanding, plots of the fire intensity resulting from the methods should be shown (possibly in an appendix if necessary). The "fuel moisture -> combustion completeness -> fire intensity -> mortality" link is a critical pathway in these results, it should be discussed explicitly but is not.

We thank the reviewer for this this comment. Unfortunately, the simulation of fire spread and fire behaviour in SPITFIRE does not have the link "fuel moisture -> combustion completeness  -> fire intensity -> mortality". Fuel consumption depends on fuel moisture in SPITFIRE, hence it is not a fixed parameter like consumption completeness, which is often used in other fire models. Surface fire intensity thus depends on the consumed fuel, fire spread and wind speed. The surface fire intensity is then used twofold: 1) to check if it is too low to support a spreading fire. If this is the case, number of fires, burnt area and all  fire effects are set to zero. 2)  Surface fire intensity is used to quantify flame length to quantify if the flame could scorch the canopy. If this is the case,  fire mortality from crown scorch is quantified and the biomass of the dead trees is distributed to the dead fuel classes (see eqs. 5-7). Please note, that the model does not simulate active crown fires. In our opinion surface fire intensity is temporarily highly variable which makes it difficult to plot it into a map and interpret its influence on fire behaviour. We hope that we could clarify the role of surface fire intensity on linking fuel moisture and tree mortality with this explanation. We regard the influence of fire spread and fire danger index as the more important variables in SPITFIRE compared to the fire intensity, because they impact fuel consumption, fire intensity, and tree mortality. With the support of Fig. 3 we discussed, why and how a changing fire danger index has an impact on burnt area. To make this point clearer we added the following text to the manuscript describing  Fig. 3 in P8 L 22-30:

*,The general behavior of the two indices as modelled by LPJmL in dependence of relative humidity and temperature is shown in Fig. 3. The Nesterov index shows a strong but very localized maximum for high temperatures and a small humidity. Hence a spreading fire is only possible in a very small climate range (here ca. from 25° Celsius and a relative humidity smaller than 0.5). The VPD on the other hand shows a less pronounced maximum but a medium fire danger also for wetter and colder regions. The slope of  towards lower VPD values is also smaller compared to the Nesterov index. Especially in regions with temperatures colder than 20°C  and relative humidity smaller than ca. 0.6 a fire is still possible.  This might increase the area in which fires can occur compared to the Nesterov index, which could be an important improvement, enabling SPITFIRE to simulate more fire in wetter and colder regions. The calculated VPD and NI values shown in Fig. 3 are based on a LPJmL-SPITFIRE run, and thus the influence of vegetation distribution on both fire danger indices.'*

6. There is no benchmarking of the PFTs that we expect to be effected by fire! The inclusion of TrBE PFT FPC is great, but what about TrBR and TrH? These should be at least plotted, and ideally benchmarked. If the ESA CCI dataset does not have useful classifications in this regard, at least MODIS VCF MOD44B Tree-Nontree-Bare would provide some reference data for the Caatinga and Cerrado.

We thank the reviewer for pointing out that this information is also important for the reader to get a complete picture of the spatial distribution of all tropical PFTs in the study region. As suggested by the Reviewer we added the benchmarking of TrBR and TrH to the Appendix and changed the text accordingly in the Results (20 L5-6) (see answer to question 27 and A1 and A2).  The maps of the PFT distributions were derived from the ESA CCI land cover map V2.0.7 (Li et al., 2018; Forkel et al., 2014).

Specific comments to the text
Abstract

7. 'partly poor'
-rephrase

We changed the sentence into:

*'However, most fire-enabled DGVMs have problems in capturing the magnitude, spatial patterns, and temporal dynamics of burnt area as observed by satellites.'*

8. 'as a starting point'
– rephrase, this is not the first work to improve fire in DGVMs

This might have been a misunderstanding. The term ‚as a starting point' refers to the

improvements related to improving the SPITFIRE model. To clarify we excluded ,as a starting point'.

9. 'improves simulation of ... plan functional type'
– is that really demonstrated?

It is demonstrated that the distribution of the PFTs improve for the fire-prone regions Caatinga and Cerrado. As suggested by the Reviewer we provide more details and clarifications also about the other PFTs (see answers to questions 27 and 30). To clarify we add here "distribution":

*'improves simulation of ... the spatial distribution of plant functional types'*

Introduction
10. P2 'recent decline in global burnt area'
– now contested (indeed by one of the authors)

We thank the Reviewer for noting this and changed the sentence into:

*'Despite a tendency for globally declining burnt area (Andela et al., 2017; Forkel et al. 2019), more frequent and [...]'*

11. P2 'Especially in South America, tropical forests, woodlands and other ecosystems are vulnerable to increasing fire danger and land use change'
– reference?

As suggested by the Reviewer we added a reference:
Cochrane, M., & Laurance, W. (2008)
[https://bioone.org/journals/AMBIO-A-Journal-of-the-Human-Environment/volume-37/issue-7/0044-7447-37.7.522/Synergisms-among-Fire-Land-Use-and-Climate-Change-in-the/10.1579/0044-7447-37.7.522.full]

Material and Methods
12. P6 'SPITFIRE further includes a surface intensity threshold'
– please state this threshold here. I realise that this is in the Table 2 but the units are not given.

We added the threshold as suggested by the Reviewer. The parameter is the fraction of burnt area per gridcell, hence has no unit. We added the threshold in P6 L28-29:

*,SPITFIRE further includes a surface intensity threshold ($10^{-6}$, fraction burnt area per grid cell), which describes the threshold of the possible area burnt[...]'*

13. P7 'The fire danger index is scaled by a PFT-dependent constant, $\alpha_i$, over the number of PFTs n(Thonicke et al., 2010)'

- in the original Thonicke et al. 2010 implementation, the α varied over fuel classes (ie. 1hr, 10hr, 100hr, 1000hr and live grass fuels) not PFTs. Please explain and justify this change. Also, there no mention of live grass fuels. Are they parameterised as in original SPITFIRE?

We thank the Reviewer for noting this. We indeed forgot to state in the paper that the fire danger index calculation of SPITFIRE had changed with the publication of LPJmL4 (Schaphoff et al. 2018) compared to the original version in Thonicke et al. (2010). The scaling over the relative moisture content of the 1-h, 10-h and 100-h fuel classes did no no longer allow a stable modelling performance in LPJmL4.0 and had therefore been replaced as an average, PFT dependent parameter (see Eq. (63) in Schaphoff et al. 2018). The calculation of the moisture content for the live fuel consumption remained the same in LPJmL4.0 as well as in this study as described in Thonicke et al., (2010), the original SPITFIRE implementation. We added the following paragraph to the methods in P7 L20-24:

*'The resulting fire danger index has been calculated as in Schaphoff et al. 2018a (slightly different compared to Thonicke et al. 2010) by taking into account the NI as measure for weather conditions and a PFT dependent scaling factor alpha_NI_i:*

$$FDI_{NI} = max\left(0, 1 - \frac{1}{m_e}exp\left(-\frac{\sum \alpha_{NI_i}}{n} \cdot NI\right)\right)$$,

*where n is the number of PFTs and m_e the moisture of extinction, which is a PFT-dependent parameter and is weighted over the litter amount.'*

14. P8 *'and a monthly mean for R to avoid unrealistic high flammability fluctuations in time steps with isolated events of very low R'*
– can the authors justify this further? I know it is stated in the Pechnoy and Shindell paper, but it is not immediately clear why flammability fluctuations due to rainfall events should be 'unrealistic'. Perhaps with their experience with this method, the authors can provide a more convincing argument.

We justified this further by adding the following paragraph in P8 L7-11:

*'The soil is a natural buffer for drought periods and heavy rainfall events. In the Nesterov index this was taken into account by the cumulative nature of this index. Since the VPD-based fire danger index is not cumulative, this buffering effect is taken into account by taking the monthly mean of the precipitation. In doing so we avoid unrealistic high flammability fluctuations in time steps with isolated events of very low or very high precipitation (R).'*

15. P8 'Hence, we scaled the VPD up with a PFT-dependent scaling factor α i '
– since this has a very different physical meaning than the α i above, I strongly suggest using a different symbol.

We followed the reviewers suggestion and changed both alphas: The scaling factor for the

Nesterov index is now called  $\alpha_{Nli}$  and the scaling factor for the VPD  $\alpha_{VPDi}$ . (See also Question 13)

16. P8 'The general behaviour of the two indices as modelled by LPJmL in dependence of relative humidity and temperature is shown in Fig. 3'
– Fig 3 is a nice plot, but please explain in a little more detail how the panels are comparable, as in how was the effect of vegetation taken in to account in the lower panel for VPD FDI?

We thank the reviewer for the useful comment and added more details on the Figure in P8 L22-30, see suggested text on our answer to main concern 5.

17. P8 'We regridded and aggregated the data set to the LPJmL resolution of 0.5 ∘ × 0.5 ∘ and to a daily time step'
– normally climate data is the limiting factor when it comes to spatial resolution in DGVMs. Is there any reason that the authors chose to aggregate this rather that use 0.25 degree? Especially when the evaluation data sets are available at 0.25 degree or finer. It seems like throwing away information.

The reviewer raised an interesting point. We decided to keep the LPJmL spatial resolution at 0.5 ∘ × 0.5 ∘ because it is also the standard resolution of other DGVMs in ISIMIP and Trendy which makes our results comparable to these models. Furthermore a gridcell size of 0.25° x 0.25° would slow down the computation time of global model runs by a factor of 4. For the optimization we also would have to use four times the amount of tested grid cells to keep the ratio of tested grid cells to untested grid cells. In our opinion, the loss of information by taking a 0.5 ∘ × 0.5 ∘ instead of a  0.25° x 0.25° gridcell size is rather small for our large scale modelling approach. We hope that the reviewer can agree to our argumentation.

18. P10 'The optimization was performed for 40 grid-cells in South America to represent a variety of fire regimes (Fig. 2). Most of them were selected in active fire regions, especially in the Cerrado and Caatinga. In addition a few pixels with no or almost no fire occurrence (e.g. central Brazilian Amazon) were chosen.'
– this is a rather vague description of what may be a very important choice in the optimisation procedure! See my main concern above. Please give more details in the logic here.

We thank the reviewer for noting this lack of explanation on how the grid cells were selected . We addressed this problem in the answer to main concern 3 and added a paragraph to the methods in P10 L22-31 (see answer to main concern 3).

19. P10 –Despite being important the FDI NI and the Rothermal equations, and being poorly

constrained, moisture of extinction was not mentioned as a possible parameter for optimisation. Could the authors discuss this?

We agree that the moisture of extinction could be also a possible parameter for the optimization. We decided not to use this parameter in order to make the optimization for both fire danger indices more comparable. Without the moisture of extinction, both fire danger indices are optimized with one PFT-dependent scaling factor. Another optimized parameter in the fire danger routine would give the Nesterov index more weight in the optimization compared to the VPD.

20. P11 'NMSE'
– can the authors justify their choice of NMSE over NME?

Generally the NMSE and the NME should show very similar results with the NMSE being the squared error term. We added a paragraph to justify better our choice of the NMSE in P12 L11-14:

'We chose the NMSE to represent and compare the model errors, as it has a squared error term, which puts a stronger emphasis on large deviations between simulations and observations as compared to a linear term, and due to its normalization it is comparable across different parameters. Especially for fire simulations we have a relatively large deviation between simulations and observations.'

21. P11 'Willmott coefficient'
– please explain its range and meaning, as is done for NMSE.

It is true that we forgot to give an overview about the different values for the Willmott coefficient. We added the following sentence inP12 L17-20:

'The Willmott coefficient is a squared index, where a value of 1 stands for perfect agreement between simulated and modelled runs and gets smaller for worse agreements with a minimum of 0. Unlike the coefficient of determination, the Willmott coefficient is additionally sensitive to biases between simulations and observations.'

Results
22. P12 'mainly by shifting much of the simulated burnt area from the sparsely vegetated Caatinga towards the Cerrado region'
– this is true to some extent, but it also much is moved into Amazonia in regions where very little fire is observed in reality. In order to back up this statement, a table with the burnt area in each region for each simulation should be provided. The overestimation of fire in Amazonia should be discussed in the Discussion section.

It is true, that much of the fire is also moved into Amazonia regions. With the Nesterov index fire was strongly underestimated in the Amazonia region, while the optimized VPD fixes this

underestimation. Fires are present at the edges of the Amazon (both in model and observation, see Fig. 4), where tree density is lower and deforestation takes place. In the closed continuous forest area towards the center of the Amazon almost no fire is observed and also not simulated.  We initially did not add the Amazonia region to Fig. 5 and Table 1, because the focus of this paper lies on the Cerrado and Caatinga region. But we agree that a comparison between all three regions is useful and added the Amazonia region to Table 1 and added a Figure of the timeseries of the Amazon to the Appendix. Furthermore we added a paragraph to the Results in P14 L4-11:

*'Also the Amazonia region mostly improved by using the VPD_optim version (Tab. 1 and Fig. A3). The R² and the Willmott coefficient improved, while the NMSE increased slightly. With the Nesterov index fire was strongly underestimated in the Amazon region, while the optimized VPD fixes this underestimation. The fire is only modelled (and also observed, see Fig. 4) at the edges to the Amazon, where wood density is lower and deforestation takes place. In the closed continuous forest area towards the center of the Amazon almost no fire is observed and neither modelled. The total burnt area increased from 0.7 million ha to 4.8 million ha (for VPD_optim) , which is now a bit overestimated to the observed burnt area of 3.4 million ha. Using the NI_optim all error metrics as well as the total burnt area decreased.'*

and to the Discussion section in P23 L8-10:

*'The poorly modelled PFT distribution also is responsible for the overestimation of the burnt area in the Amazon region. Because of the too large fraction of TrBE in the Cerrado/Caatinga region the scaling factor for this PFT is relatively high. This leads in turn to an overestimation in the Amazon region, where the fraction of the TrBE is larger.'*

| Region | NMSE | Willmott | Sum |
|---|---|---|---|
| **Spatial - South America** | | | |
| GFED | | | $1.9 \cdot 10^7$ |
| $NI_{orig}$ | 1.80 | 0.27 | $1.4 \cdot 10^7$ |
| $VPD_{optim}$ | 0.82 | 0.56 | $1.6 \cdot 10^7$ |
| $NI_{optim}$ | 1.09 | 0.08 | $0.2 \cdot 10^7$ |
| **Temporal - Cerrado** | | | |
| GFED | | | $9.2 \cdot 10^6$ |
| $NI_{orig}$ | 0.30 | 0.89 | $5.2 \cdot 10^6$ |
| $VPD_{optim}$ | 0.27 | 0.90 | $6.4 \cdot 10^6$ |
| $NI_{optim}$ | 1.07 | 0.36 | $0.6 \cdot 10^6$ |
| **Temporal - Caatinga** | | | |
| GFED | | | $0.4 \cdot 10^6$ |
| $NI_{orig}$ | 327.82 | 0.14 | $6.0 \cdot 10^6$ |
| $VPD_{optim}$ | 15.2 | 0.46 | $1.6 \cdot 10^6$ |
| $NI_{optim}$ | 1.07 | 0.73 | $0.3 \cdot 10^6$ |
| **Temporal - Amazonia** | | | |
| GFED | | | $3.4 \cdot 10^6$ |
| $NI_{orig}$ | 0.83 | 0.56 | $0.7 \cdot 10^6$ |
| $VPD_{optim}$ | 0.93 | 0.83 | $4.8 \cdot 10^6$ |
| $NI_{optim}$ | 1.22 | 0.32 | $0.02 \cdot 10^6$ |

Tab. 1: Comparison of the results in terms of NMSE, the Willmott coefficient of agreement and the sum (in ha per year) between NI_orig, VPD_optim, NI_optim and the GFED evaluation data

[Figure]

Fig. A6. Time-series of monthly burnt area from 2005 - 2015 simulated by SPITFIRE (red lines) compared to GFED4 evaluation data (blue lines) for: (a) The Amazonia region, using NI_orig. (b) Total South America, using the NI_orig. (c) The Amazonia region, using NI_optim. (d) Total South America, using NI_optim. (e) The Amazonia region, using VPD_optim. (f) Total South America, using VPD_optim.

23. P13 Figure 5
- There is some fire in the Amazonia region, both in the data and in the simulations. Therefore, this region should be included in Figure 5 and Table 1, and discussed.

We now added the Amazonia region to Table 1 and the timeseries of the region to the Appendix. Please see more discussion to this point in question 22.

24. P13 'Here, the TrBE showed the largest value (22.41), ca. 20 times as large as the TrBR (1.21) and TrH (1.13) (Tab. 2)'

– there is no discussion of what this actually means in the Discussions section, please include an interpretation.

We agree that the new alpha values deserve some further discussions and added a paragraph to the Discussion section in P22 L2-12 (see main concern 4).

25. P17 '... but also here we got an even larger improvement, when only the fire-prone regions Cerrado or Caatinga are considered (Tab. 3)'
- Caatinga results are not shown in Tab. 3, although I think they should be. Possibly also results for Amazonia (see above)

We focused in this Table on showing the larger improvements of fire in the fire prone Cerrado (where the fire optimization has a large impact), in comparison to the only small improvements in total South America. We did not show other regions and PFTs, because this would enlarge the Table/Figures and the Result/Discussion part and go beyond the scope of this paper. We see the usefulness of further regions and PFTs and now added the Caatinga to Table 3. We decided to not add the Amazon region in this context, because the main message of this table is, that in the fire-prone regions (Cerrado and Caatinga) the improvements in model performance are much larger compared tototal SA (which has several areas without fire).
We however now included Amazonia in Tab 1 and Fig. A3 (see answers to questions 22 and 23). We hope that this additional information now describes our results better.

| Region | NMSE | Willmott |
|---|---|---|
| **AGB** | | |
| SA ($NI_{orig}$) | 0.97 | 0.83 |
| SA ($VPD_{optim}$) | 0.91 | 0.84 |
| SA ($NI_{optim}$) | 0.99 | 0.83 |
| Cerrado ($NI_{orig}$) | 15.06 | 0.25 |
| Cerrado ($VPD_{optim}$) | 12.36 | 0.28 |
| Cerrado ($NI_{optim}$) | 16.06 | 0.24 |
| Caatinga ($NI_{orig}$) | 11.93 | 0.32 |
| Caatinga ($VPD_{optim}$) | 8.57 | 0.36 |
| Caatinga ($NI_{optim}$) | 10.44 | 0.33 |
| **FPC - Evergreen (TrBE)** | | |
| SA ($NI_{orig}$) | 0.42 | 0.82 |
| SA ($VPD_{optim}$) | 0.41 | 0.82 |
| SA ($NI_{optim}$) | 0.43 | 0.81 |
| Cerrado ($NI_{orig}$) | 1.04 | 0.60 |
| Cerrado ($VPD_{optim}$) | 0.70 | 0.64 |
| Cerrado ($NI_{optim}$) | 1.40 | 0.55 |
| Caatinga ($NI_{orig}$) | 1.73 | 0.40 |
| Caatinga ($VPD_{optim}$) | 1.54 | 0.29 |
| Caatinga ($NI_{optim}$) | 2.05 | 0.44 |

Tab. 3. Comparison of the results for AGB and the TrBE PFT cover in terms of NMSE and the Willmott coefficient of agreement between NI_orig, VPD_optim and NI_optim in South America (SA), in the Cerrado and in the Caatinga.

26. P 18 Figure 7
– difference plots are great and I can see the logic behind including the difference relative to the original model version to show improvements (as you have done) but please show the absolute values too (as in Figure 4).

We decided to not show the absolute values, because it is not possible to see any difference between the original model versions by eye. Hence, a difference plot is the best way to show the changes between the different model versions. We added the figure for the absolute values to the Appendix (Fig. A5).

[Figure]

Fig. A5. Annual above ground biomass (AGB) of trees over a mean from 2005-2015 in kgC/m^2. (a) Avitabile evaluation data. (b) Simulated AGB by LPJmL4-SPITFIRE in the NI_orig version. (c) Simulated AGB by LPJmL4-SPITFIRE in the VPD_optim version. NI$_orig. (d) Simulated AGB by LPJmL4-SPITFIRE in the NI_optim version.

Discussion

27. P19 'Another result of the optimizing procedure, using FDI_VPD , was the improvement of the PFT distribution..'

– I am not sure that statement is justified given the very small improvement in TrBE and no demonstrated improvement in the other PFTs.

We agree that the improvement of the PFT distribution in total South America is relatively small. But since we only optimize fire parameters, we can only improve areas, where fire was misrepresented in SPITFIRE. Here an optimized burnt area can have a feedback on the PFT distribution. As demonstrated in the fire-prone Cerrado the improvement of PFT distribution is relatively large (the NMSE is halved compared to the original model version). Since we have in most parts of northern South America only two tree PFTs (TrBE and TrBR) and a very small amount of grass, we did just show the TrBE, assuming that the TrBR improves accordingly. Now we show (and discuss) the distribution of the other PFTs in the supplement.

It is the aim of our paper to show the improvement of burnt area due to the new fire danger index and the optimization. Hence, it was important that the improved fire representation does not decrease the performance of the PFT distribution. For the fire-prone regions, with large fire-vegetation feedbacks, we even find an improved PFT distribution.

[Figure]

Fig A1. Annual FPC cover by tropical broadleaved raingreen PFT over a mean from

2005-2015 as fraction per cell. (a) ESA-CCI evaluation data (b) Simulated FPC by LPJmL4-SPITFIRE using the NI orig version (c) Simulated FPC by LPJmL4-SPITFIRE using the VPD optim version (d) Simulated FPC by LPJmL4-SPITFIRE using the NI optim version

[Figure]

Fig. A2. Annual FPC cover by tropical herbaceous PFT over a mean from 2005-2015 as fraction per cell. (a) ESA-CCI evaluation data (b) Simulated FPC by LPJmL4-SPITFIRE using the NI orig version (c) Simulated FPC by LPJmL4-SPITFIRE using the VPD optim version (d) Simulated FPC by LPJmL4-SPITFIRE using the NI optim version

We added a short description to the result section in P20 L5-6:

*Also for the the TrBR and TrH PFT distributions we got an improved performance using the VPD_optim and mostly a worse performance using the NI_optim version (Fig. A1 and A2).*

28. P19 'it emphasizes that three parameter sets determining PFT distribution'
– what three parameter sets? You mean three PFTs? Or something else? Please clarify.

We agree with the Reviewer that this sentence is not clear. We rewrote it as following in P21 L25-27:

*'Hence, it emphasizes that we need to include further parameters in the optimization, which impact directly the PFT distribution, biomass and fire to obtain a significant improvement in the spatial and temporal distribution of both, vegetation and fire. However, this study focused solely on the parameters within the SPITFIRE module.'*

29. P20 'Limitations during the optimization process'
– this heading is somewhat confusing and maybe should better be 'Limitations of the optimization process'

We thank the Reviewer for this suggestion and changed the title of the subsection accordingly.

30. P20 'As shown in Fig. 8, the modelled PFT coverage showed an equal distribution of tropical raingreen and evergreen PFTs throughout wide parts of central-northern South America'
– Fig. 8 shows no such thing, it only shows the FPC of the evergreen PFT. Of course, it may simply be that the caption is incorrect somehow, but otherwise the distribution of the raingreen PFT must be shown to demonstrate this.

We now show the fraction of the raingreen and herbaceous PFT in the Appendix (see answer to question 27). We added this information to the text in P23 L2-3:

*'As shown in Fig. 8, A1 and A2, the modelled PFT coverage showed an equal distribution of tropical raingreen and evergreen PFTs throughout wide parts of central-northern South America'*

31. P20 'By choosing a large amount of optimization cells in the, by NI orig , strongly overestimated Caatinga region, the burned area decreased there significantly after the optimization'
– this (slightly confusing statement) would appear to indicate that the authors acknowledge that their results depend heavily on the choice of grid cells for the optimisation (see above)

We thank the Reviewer for noting that we could discuss this point better. We addressed the cell selection method in the answer to main concern 3. Some further issues with the optimization procedure for the different fire danger indices and regions were discussed in the answer to main concern 1. As stated in the answer to main concern 3 we changed a part of the Discussion in P10 L22-31.

32. P20 'In the Cerrado and especially the Caatinga, however, trees suffer from water stress in the dry season and should shed their leaves to avoid mortality related to drought or growth efficiency. The resulting dominance of the TrBR PFT has a very different effect on fire spread and is more fire-tolerant (different fuel characteristics and resulting fire intensity), thus has a lower fire-related mortality.'
– whilst this a reasonable enough statement (in fact pretty much inherent in the construction of DGVMs and SPITFIRE) it is hard to see what it has to do with the limitations of the optimisations process.

The aim of this statement was to emphasize the importance of a correct PFT distribution. If we have ca. 50% raingreen PFTs in the Amazon (which should rather establish in the drier Cerrado and Caatinga region) their specific traits pose a problem to the optimization procedure. But we agree that this sentence could be omitted. Hence, we deleted this sentence and changed the paragraph (P23 L5-8) into:

*'In the tropical rainforest the TrBR proportion is overestimated, which leads to problems in the optimization procedure, since TrBR has very different effects on fire spread and is more fire-tolerant (different fuel characteristics and resulting fire intensity). This leads to a lower fire-related mortality, which fits better to the drier and fire prone savanna-like regions (e.g. Cerrado).'*

33. P21 – 'Nonetheless, we were able to improve the interannual variability and hence, the model performance during extreme years for the Cerrado and Caatinga regions (e.g. for 2007/2008, Fig. 5). The optimized SPITFIRE is now able to model accurately the climate dependent seasonal and interannual variability as well as the spatial extent of fire on natural land throughout the fire-prone woodlands of South America.'
– yes and no. In the Cerrado the results from Fig 5. are not significantly different between VPD and Original, and whilst the results are better in the Caatinga for VPD, most of this comes down to the overall normalisation, it is hard to see if VPD really catches between IAV and seasonal dynamics. In fact, the $R^2$ (which is insensitive to the normalisation) actually gets worse going from Original to VPD. So these statements need much more nuance. And a plot of the normalised time series (equivalent to Fig 5., at least for the Caatinga) might be a more effective way showing improvements in IAV and seasonal dynamics.

We thank the Reviewer for noting that we could discuss more about the raised concern. We addressed this issue in the answer to main concern 1. Furthermore, thanks to the reviewer's comment we noted a small error in Figure 5: The $R^2$ of 5a (original model version for the Cerrado) has been wrongly written as 0.87. The real value, which was correctly written in the original text of the previous manuscript version, is however 0.78. Hence we have a slight improvement by the VPD version in all three metrics. We are very sorry for the confusion this spelling error has caused.

In the revised manuscript we now also show a Figure of the Amazonia region and for total South America in the Appendix (see Fig. R3). In total South America all the metrics improve significantly which indicates an improved IAV. While we have a large improvement in the Caatinga and in total South America, the performance and IAV for Amazonia and the Cerrado region improved slightly. The spatial extent gets better for the whole area as shown in Fig 4.

We also included now the p-value, which is mostly smaller using the VPD, indicating a more significant correlation. To make our statement clearer we rewrote the paragraph into (P23 L21-25):

*'Nonetheless, we were able to improve the interannual variability and hence, the model performance to a great extent for the Caatinga and slightly for the Cerrado and Amazon regions (Fg. 5 and A3). The Cerrado already had a very good modelling performance before the optimization process, which now only slightly improved. The performance of the interannual and seasonal variability of burnt area for total South America improved substantially (Fig. A3). The optimized SPITFIRE is now better able to simulate accurately the climate dependent seasonal and interannual variability as well as the spatial extent of fire on natural land throughout the fire-prone woodlands of South America.'*

We also agree with the raised concern about a more effective way to show improvements in IAV and seasonal dynamics for the Caatinga. We tried different normalization approaches, which did not lead to a better visualization. In the end we decided to show a logarithmic scale for the Caatinga in order to take into account the large differences between observation and the different model versions. A version of the plots with a non logarithmic scale remains in the Appendix.

[Figure]

Fig. 5. Timeseries of monthly burnt area from 2005 - 2015 simulated by SPITFIRE (red lines) compared to GFED4 evaluation data (blue lines) for: (a) The Cerrado region, using NI_orig. (b) The Caatinga region, using the NI_orig. (c) The Cerrado region, using NI_optim. (d) The Caatinga region, using NI_optim. (e) The Cerrado region, using VPD_optim. (f) The Caatinga region, using VPD_optim. Note the logarithmic scale for the Caatinga, which was applied in order to account for the large differences between the different model versions (for a non logarithmic version see Fig. A6).

34. P21 entire section titled 'Outlook
- the way ahead in improving fire modules in DGVMs' – this text does not really fit the title. Much of it refers specifically SPITFIRE or LPJml, specifically their current limitations. Please reconsider/revise/re-title this section.

We thank the Reviewer and followed the suggestion and retitled this section. The name is now: *Limitations of fire modeLling in LPJmL4-SPITFIRE.*

35. P21 The statements 'it would be possible to use an even more comprehensive fire

danger index (e.g. Canadian Fire Weather Index; Wagner et al., 1987) or different fire danger indices for different biomes' and 'In a global modelling approach, however, we need to find one fire danger index' seem to contradict each other, please resolve!

It is true that both statements contradict each other. We wanted to express, that it would be ideal to have different fire danger indices for different biomes, because of the different fire dynamics. However, in our global modelling approach it is necessary to have just one fire danger index. The reason for this is, e.g., the computational effort or the exact definitions of biomes, which are also changing within one model run and hence do not fit to the LPJmL model logic. We have now removed the first sentence to avoid the confusion and changed the next sentence slightly in P24 L17-19:

*'Fire models embedded in DGVMs should build on a FDI which is complex enough to account for various fire dynamics, while it's parameterization should be simple enough to be accurately applied on a global scale. While the VPD is more complex and takes into account more climatic input as the Nesterov index, it is still relatively easy to implement in a global fire model.'*

Conclusions

36. P21 'We have demonstrated a major improvement of the fire representation within LPJmL4-SPITFIRE by implementing a new fire danger index and applying a model-data integration setup to optimize fire-related parameters.'
- whilst there are tangible improvements, they are only tested and in the Caatinga and Cerrado, the region for which the optimisation was done (which you do mention in the next sentence). I would suggest toning this down slightly.

We are confident that the revised manuscript now shows better the improvements in performance of modelled burnt area in South America. As suggested by the reviewer we added the study region to this sentence.

*'We have significantly improved the fire representation within LPJmL4-SPITFIRE, applied for South America, by implementing a new fire danger index and applying a model-data integration setup to optimize fire-related parameters.'*

37. P21 'We improved the seasonal and interannual variability'
– I have yet to be convinced of this, especially as the $R^2$ for the time series are not improved with VPD. And I am not sure how to interpret the Willmott coefficient as this is not described.

We addressed this issue in the answer to main concern 1 and question 33 and describe now the Willmott coefficient (see answer to question 21).
We hope to have convinced the reviewer that the revised manuscript now shows better the improvements in performance of modelled burnt area in South America.
In the revised manuscript we now also show a Figure of the time-series of total South

America and the Amazon region in the Appendix, which show the improvements in model performance and IAV of the burnt area when using the optimized VPD. Hence a large improvement in the Caatinga and total South America and a slightly better performance in the Amazon and Cerrado region leads to a total improvement of IAV. Regarding the modelled R² please see our response to question 33.

38. P21 'A realistic representation of fire is also crucial for fire-vegetation-climate feedbacks and is hence necessary for DGVMs coupled within and comprehensive Earth system model.' – I think you can drop that sentence, as it attempts to summarise and justify fire modelling in general rather than this work. The penultimate sentence is fine to end with.

We thank the Reviewer for this suggestion and dropped the last sentence.

We thank the reviewer for the detailed and focused review of our manuscript.

[revised manuscript text omitted]